# Adiponectin-expressing Treg facilitate T lymphocyte development in thymic nurse cell complexes

Yiwei Zhang[1,2,5], Handi Cao[1,2,5], Jie Chen[1,2,4], Yuanxin Li[1,2], Aimin Xu [1,2,3] & Yu Wang [1,2✉]

Adiponectin is a well-known insulin sensitizer and anti-inflammatory molecule, possessing therapeutic potentials in cardiovascular, metabolic and cancer diseases. Results of the present study demonstrate that adiponectin is expressed in a population of regulatory T-cells (Treg) resided within the thymic nurse cell (TNC) complexes. Adoptive transfer of adiponectin-expressing Treg precursors effectively attenuated obesity, improved glucose and insulin tolerance, prevented fatty liver injuries in wild-type mice fed a high-fat diet, and significantly inhibited breast cancer development in MMTV-PyVT transgenic mice. Within the TNC complexes, locally produced adiponectin bound to and regulated the expression as well as the distribution of CD100, a transmembrane lymphocyte semaphorin, in turn modulating the lymphoepithelial interactions to facilitate T-cell development and maturation. In summary, adiponectin plays an important role in the selection and development of T lymphocytes within the TNC complexes. Adiponectin-expressing Treg represent a promising candidate for adoptive cell immunotherapy against obesity-related metabolic and cancer diseases.

[1] The State Key Laboratory of Pharmaceutical Biotechnology, The University of Hong Kong, Hong Kong SAR, China. [2] Department of Pharmacology and Pharmacy, LKS Faculty of Medicine, The University of Hong Kong, Hong Kong SAR, China. [3] Department of Medicine, LKS Faculty of Medicine, The University of Hong Kong, Hong Kong SAR, China. [4] Present address: Henry Fok College of Biology and Agriculture, Shaoguan University, Shaoguan, Guangdong, China. [5] These authors contributed equally: Yiwei Zhang, Handi Cao. ✉email: yuwanghk@hku.hk

Adiponectin is a well-known circulating glyco-hormone regulating energy metabolism and immune homeostasis[1]. It possesses potent anti-inflammatory, anti-diabetic, and anti-tumorigenic activities[2–4]. Adiponectin was discovered originally in adipocytes and shares homology with type VIII and X collagens, complement factor C1q and tumor necrosis factor α (TNFα)[5–7]. Adiponectin protein forms trimers, hexamers and high-molecular-weight (HMW) species with distinct biological activities[8,9]. The circulating levels of adiponectin are inversely correlated with many cardiometabolic abnormalities and cancer diseases[10,11]. Mice lacking the alleles of ADIPOQ are more susceptible to the development of obesity-related metabolic and malignant diseases, whereas replenishment of adiponectin decreases glucose production, restores insulin sensitivity, reduces visceral adiposity, protects against hepatic steatosis and inflammatory liver injuries, attenuates the development of atherosclerotic vascular disease, and inhibits cancer development[12–18].

Apart from adipose tissue, adiponectin has been identified as a factor produced from a subset of unstimulated non-B non-T lymphocytes to inhibit granulopoiesis[19,20]. High levels of adiponectin are detected in the bone marrow, certain lymphoid cell lines, and immune effector cells purified from healthy donors[21]. Adiponectin is expressed by components of the hemopoietic stem cell (HSC) niche to increase proliferation, while retaining the HSC in a functionally immature state[22]. In the presence of stromal cells, adiponectin selectively inhibits lymphopoiesis in long-term cultures of bone marrow or those initiated with lymphocyte precursors[23]. These evidence indicate that adiponectin plays a role in regulating the survival, differentiation, or function of lymphocyte precursors. However, further characterization of the non-adipocyte source of adiponectin has been difficult due largely to the low number/abundance of the specific lineage subset.

Thymus is a major organ for the development and maturation of T lymphocytes[24]. The lymphoid progenitors from the bone marrow enter the thymus and expand by forming the double-negative (DN), double-positive (DP), and single-positive (SP) T-cells[25]. The microenvironment of thymus facilitates the commitment of lymphoid progenitors into T lineage, the positive/negative selection of newly generated T lymphocytes and the production of regulatory T-cells (Treg) for establishing self-tolerance[26]. The present study demonstrates that adiponectin is expressed in a subpopulation of progenitors that are able to develop into the mature thymic Treg (tTreg). The adiponectin-expressing tTreg are involved in the selection and development of T lymphocytes in the thymic nurse cell (TNC) complexes, in turn modulating the systemic T-cell homeostasis. Deficiency of adiponectin alters the maturation of Treg and the selection of T lymphocytes in thymus, thus facilitating the development of diseases such as breast cancer and obesity-related metabolic complications.

## Results

**Adiponectin is expressed in thymus.** Adiponectin protein was detected in the thymus of wild-type (WT) mice and existed as trimer, hexamer, and high molecular weight (HMW) oligomers (Fig. 1a). The protein concentration of adiponectin was $1.2576 \pm 0.1417\,\mu g/mg$ and $0.0065 \pm 0.0015\,\mu g/mg$ in epididymal adipose and thymus tissues, respectively, as measured by an in-house ELISA. Immunofluorescence analyses revealed that adiponectin protein was present across the outer cortex, the cortico-medullar, and medullar regions, but not co-localized with that of CD31, which labels endothelial cells of arteries, veins, and capillaries (Fig. 1b). At the cortico-medullar and medullar regions, the

signals of adiponectin protein were either co-localized or close to those of cytokeratin 5 and/or cytokeratin 8 (Fig. 1b).

The full-length mRNA transcript of ADIPOQ was detected in both epididymal adipose tissue and thymus (Supplementary Fig. 1a). In cells isolated from thymus, the ADIPOQ transcript was present in CD4$^+$ single-positive (SP), CD4$^+$CD8$^+$ double-positive (DP) and CD4$^-$CD8$^-$ double-negative (DN) subpopulations (Supplementary Fig. 1b). In DN1 cells, the ADIPOQ transcript was mainly expressed by DN1a and DN1b subsets (Supplementary Fig. 1c), which represent the T-lineage progenitors[27]. In situ hybridization revealed that most of the cells containing the mRNA transcript of ADIPOQ were located within the lymphoepithelial cell clusters (Fig. 1c), which were positively stained with antibodies against cytokeratin 5 and/or cytokeratin 8 (Fig. 1d), markers of the thymic nurse cell (TNC) complexes[28]. In contrast to the ADIPOQ transcript, which was present in only a few individual cells (Fig. 1c), adiponectin protein was widely distributed in the extracellular space between the thymocytes and the cage-like structures formed by the epithelial plasma membrane of the TNC complexes (Fig. 1d).

In thymus of the transgenic two-color reporter mouse model (Adn-Cre/ROSA$^{mT/mG}$), the expression of Cre recombinase driven by the ADIPOQ promoter resulted in permanent, stable, and highly specific EGFP (mG) signals, replacing the cell membrane-localized tdTomato (mT) fluorescence in the thymus tissue (Fig. 2a). Flow cytometry was performed to examine the nature of EGFP$^+$ cells in the thymus of Adn-Cre/ROSA$^{mT/mG}$ mice. Around 0.012% of the total thymocytes were EGFP$^+$ cells, which were labeled positively with CD45 [a lymphohematopoietic surface antigen] and negatively with CD326 [an epithelial adhesion molecule], distributed in CD4$^+$ single-positive (SP) and CD4$^+$CD8$^+$ double-positive (DP) subpopulations, with no, low or high CD3 expressions (Fig. 2b).

In cell suspension prepared from the thymus of Adn-Cre/ROSA$^{mT/mG}$ mice, most of the EGFP$^+$ cells resided within the lymphoepithelial TNC complexes (Fig. 2c), where many lymphocytes labeled with mT fluorescence were actively engulfed and released (Supplementary Movie 1). In samples containing enriched TNC complexes, approximately 0.034% of the total number of thymocytes within TNC complexes were EGFP$^+$ and distributed in CD4$^+$ SP and CD4$^+$CD8$^+$ DP subpopulations (Fig. 2d). About half of the EGFP$^+$ within the TNC complexes were positively labeled with markers of the canonical T-lineage precursors, CD117/cKit [the stem cell factor receptor] and CD25 [the α chain of the interleukin (IL)-2 receptor][27]. Moreover, ~18% of EGFP$^+$CD4$^+$CD8$^-$ and ~79% of EGFP$^+$CD4$^+$CD8$^+$ cells within the TNC complexes were CD25$^+$Foxp3$^+$ (Fig. 2d), the characteristic feature of regulatory T-cells (Treg)[29].

In samples containing enriched TNC complexes, the amount of Adipoq mRNA transcript was ~four-fold higher than that of the thymus tissue (Supplementary Fig. 2a). All three oligomers of adiponectin were present in the enriched TNC complexes (Supplementary Fig. 2b). The ADIPOQ transcript and adiponectin protein expression were both detected in EGFP$^+$ but not EGFP$^-$ thymocytes (Supplementary Fig. 2c).

**Adiponectin-expressing thymic regulatory T-cells (tTreg).** Adiponectin-expressing EGFP$^+$ cells collected from the thymus of 5-week-old Adn-Cre/ROSA$^{mT/mG}$ mice were adoptively transferred (30000 EGFP$^+$ cells/*mouse* by tail vein injection) into sub-lethally irradiated WT or adiponectin knockout (AKO) mice. Thymus, liver, and adipose tissues were collected from the recipient mice at different time points for subsequent analyses. Within 12 h after the injection, the majority of EGFP$^+$ cells were

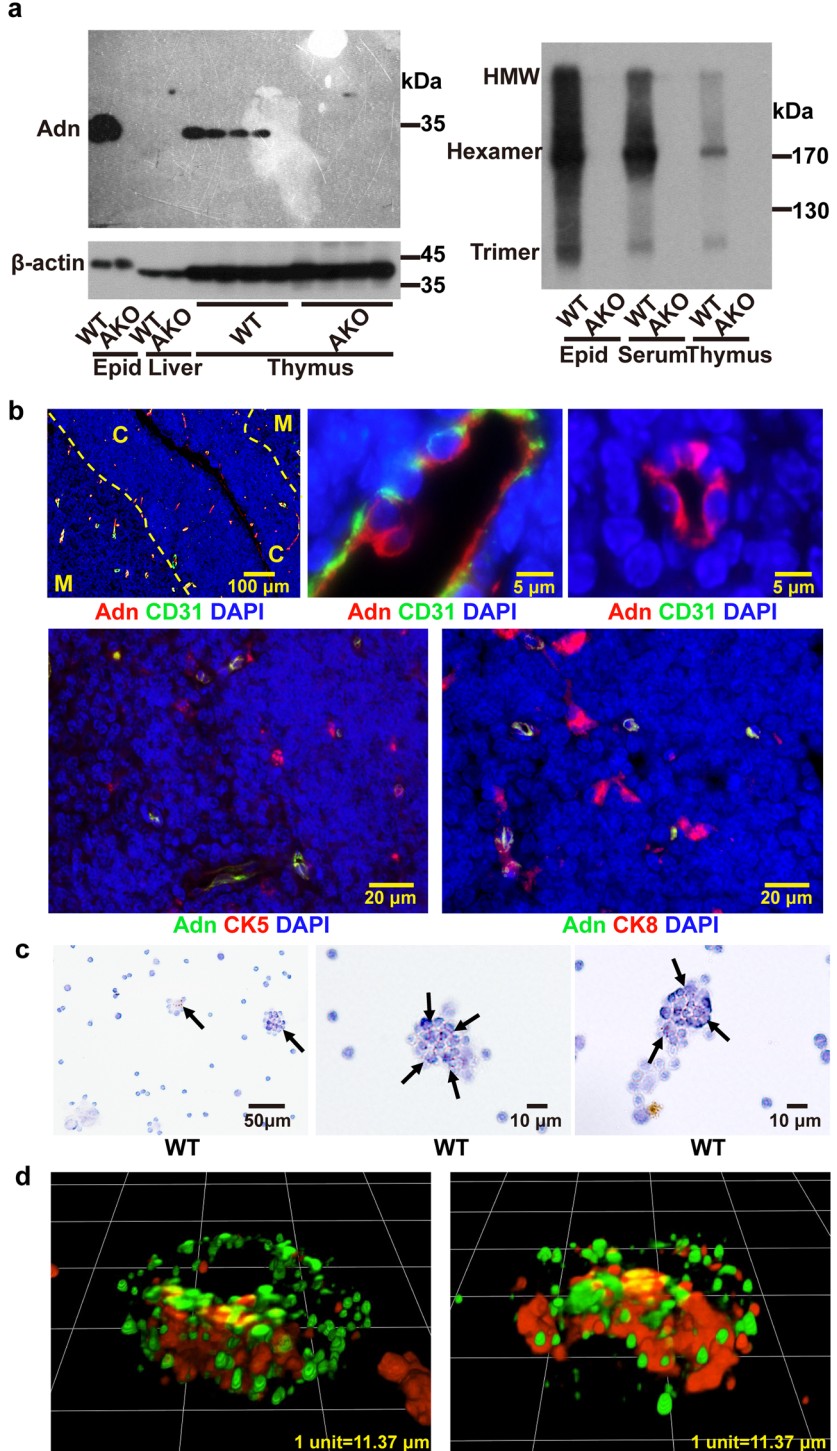

**Fig. 1 Adiponectin is expressed in thymus. a** Wild-type [WT] or adiponectin knockout [AKO] mice were sacrificed at the age of 7 weeks to collect epididymal adipose tissue [epid], liver, and thymus. Adiponectin [Adn] protein expression was analyzed by denatured (*left*) or non-reducing (*right*) SDS-PAGE and detected by Western blotting using a polyclonal antibody recognizing murine adiponectin. Beta-actin (β-actin) was probed as loading controls. **b** Immunofluorescence staining was performed to examine the expression and distribution of Adn protein in the thymus of WT mice. The tissue sections were counterstained for CD31, cytokeratin [CK] 5, CK8 and DAPI. *C* and *M* indicate cortex and medulla, respectively. **c** In situ hybridization was performed for detecting the mRNA transcripts of *ADIPOQ* in cell suspensions isolated from the thymus of WT mice. Positive brown signals were indicated by black arrows. **d** Confocal fluorescence microscopy was applied to analyze the protein expression and distribution of Adn (green) in TNC complexes isolated from WT thymus. The sections were counterstained (red) for CK5 and CK8.

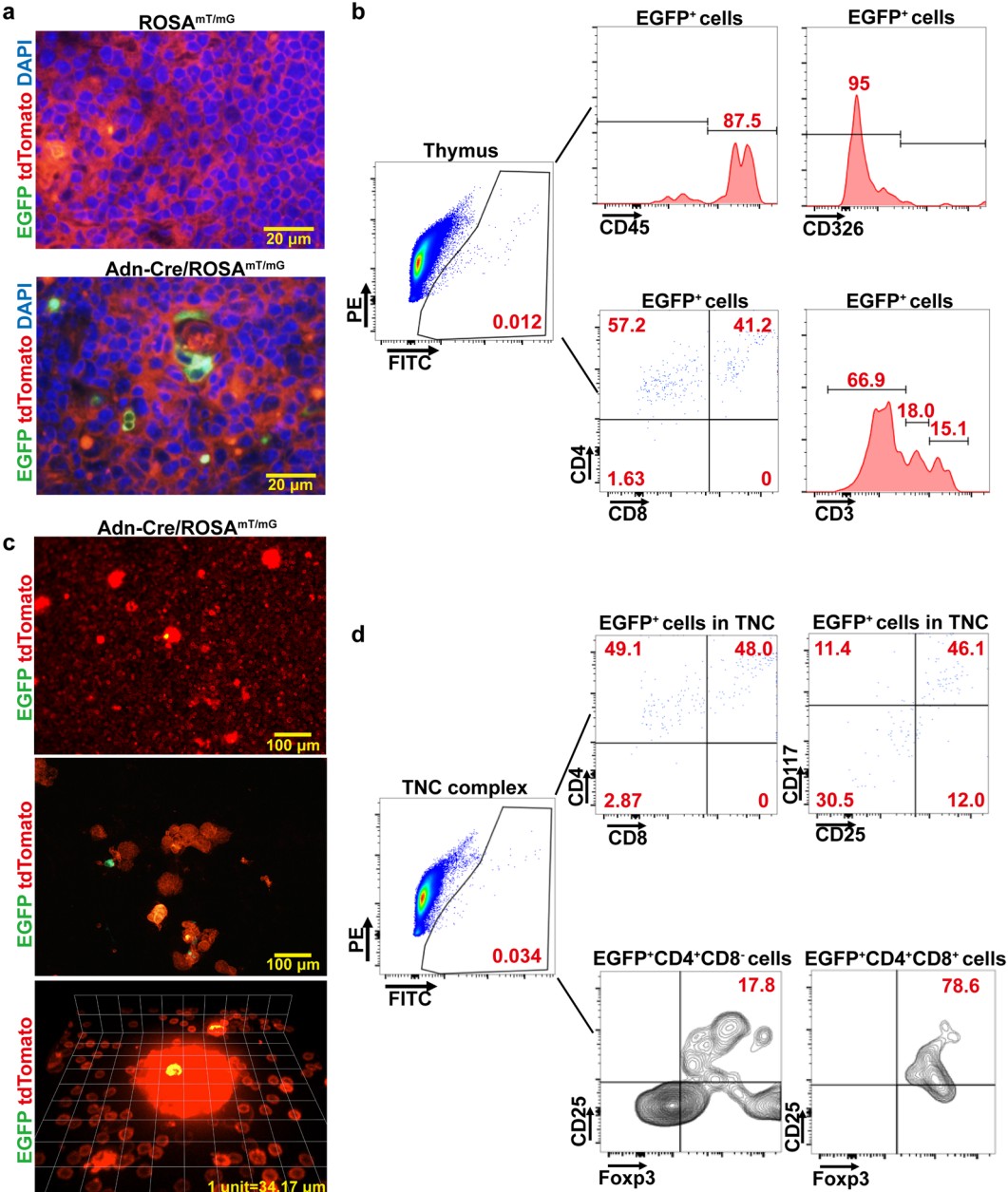

**Fig. 2 Flow cytometric analyses of adiponectin-expressing thymocytes. a** Tissue cryosections were prepared from thymus of ROSA^mT/mG and Adn-Cre/ROSA^mT/mG mice to visualize the EGFP+ cells exhibiting bright green fluorescence. **b** Single-cell suspensions were prepared from the thymus of Adn-Cre/ROSA^mT/mG mice. Flow cytometry was performed to analyze EGFP+ cells and their surface expression of markers including CD45, CD326, CD4, CD8, and CD3. **c** Cell suspensions were prepared from the thymus of Adn-Cre/ROSA^mT/mG mice (*top*). TNC complexes were then enriched by sedimentation in FBS (*middle*). The EGFP+ cells were visualized by real-time live confocal microscopy (*bottom*) [more details in Supplementary Movie 1]. **d** Flow cytometry was performed to analyze adiponectin-expressing EGFP+ cells in TNC complexes, and their surface expression of CD4, CD8, CD117, CD25, and Foxp3.

present within the TNC complexes isolated from thymus tissues of both WT and AKO recipient mice (Fig. 3a, *left*). In TNC complexes of WT recipient mice, adiponectin protein was widely distributed around the engulfed thymocytes, including the EGFP+ cells. By contrast, only a few adiponectin protein signals were located in close proximity to the EGFP+ cells within the TNC complexes of AKO recipient animals (Fig. 3a, *right*). There were barely any EGFP+ cells in the liver and epididymal adipose tissues of WT or AKO recipient mice, at one- and three-days after the adoptive transfer (Supplementary Fig. 3).

The lineage (Lin)-negative thymocytes were collected from the thymus of WT and AKO recipient mice for flow cytometric analyses of donor-derived EGFP+ cells. On the first day after injection, the majority of EGFP+ were CD117+CD25+ and CD4+, with less than 10% exhibiting CD4+CD8+ in the thymus of both WT and AKO recipient mice (Fig. 3b). On the 15th day after injection, approximately 25 and 50% of EGFP+ cells were CD4+CD8+ in WT and AKO thymus, respectively (Fig. 3c). Note that over 37% of EGFP+CD4+CD8− cells in WT thymus exhibited CD4+CD25+Foxp3+ (Fig. 3c), thus developed into thymic Treg (tTreg)[29]. Compared to WT, the percentage content and the number of EGFP+ cells in the thymus of AKO were significantly decreased at day 15 after adoptive transfer (Fig. 3d). As a result, the total number of the mature EGFP+ Treg in AKO thymus was significantly less than those of WT recipient mice ($3790 \pm 285$ vs $46968 \pm 2225$, $P < 0.05$). On the 15th day after

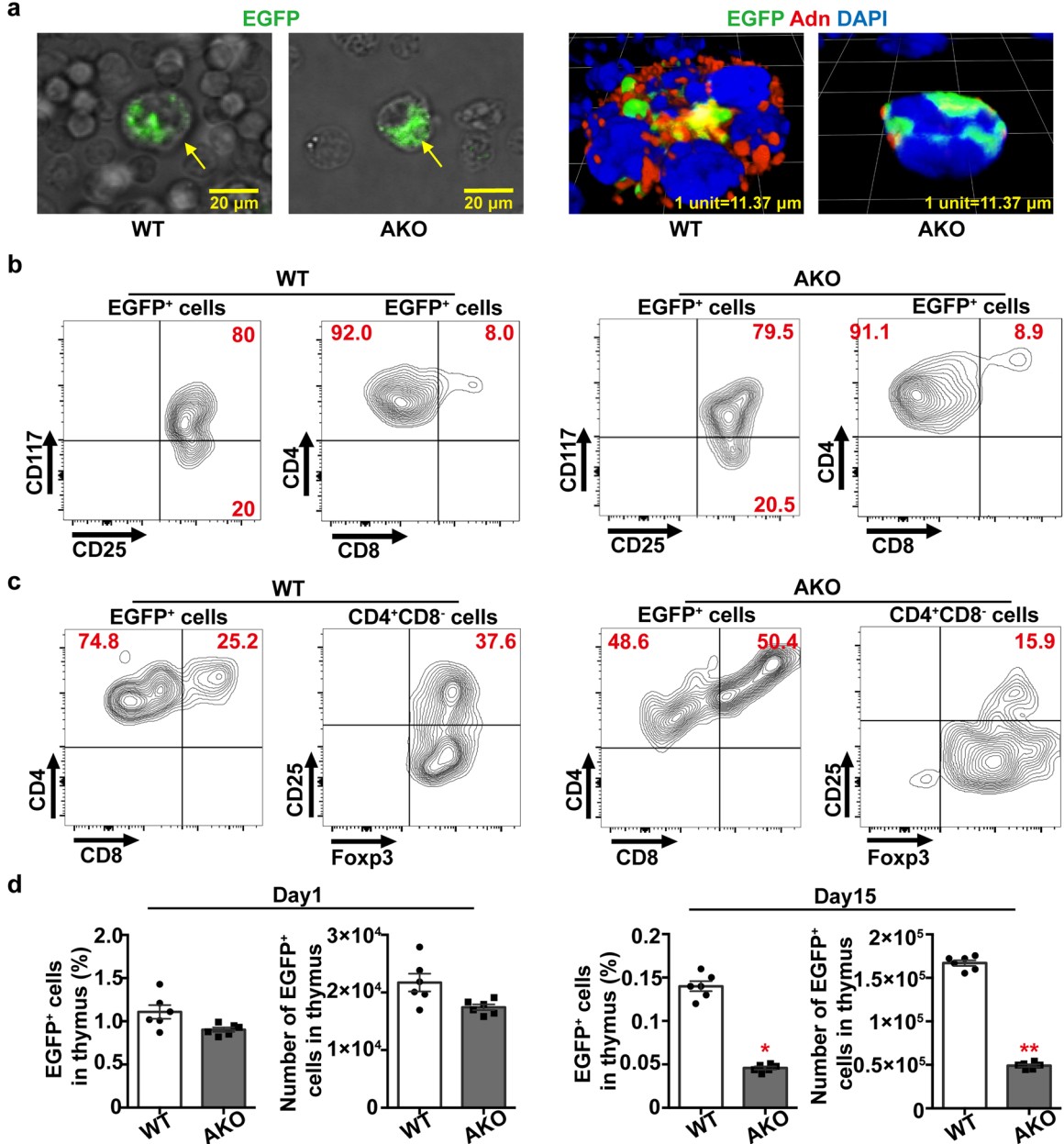

**Fig. 3 Adiponectin-expressing thymocytes are tTreg precursors. a** EGFP⁺ cells collected from the thymus of Adn-Cre/ROSA^mT/mG mice were injected [30000 cells/*mouse* via tail vein] into WT or AKO recipient mice, which were subjected to a sub-lethal 5 Gy γ-radiation. At 12 h after adoptive transfer, cell suspensions were prepared from the thymus of the recipient mice for visualizing the EGFP⁺ cells (*left*) and immunofluorescence staining for adiponectin protein (*right*). **b** At one day after cell injection, flow cytometry was performed to analyze the distribution of EGFP⁺ cells in the thymus of WT (*left*) and AKO (*right*) recipient mice, after staining with antibodies recognizing CD117, CD25, CD4, and CD8. **c** At 15 days after injection, flow cytometry was performed to analyze the distribution of EGFP⁺ cells in the thymus of WT (*left*) and AKO (*right*) recipient mice, after staining with antibodies recognizing CD4, CD8, CD25, and Foxp3. **d** The percentage contents and total numbers of EGFP⁺ cells in the thymus of WT or AKO recipient mice were calculated based on flow cytometric results at day one and 15 after adoptive transfer. Data are presented as mean ± SEM. *, $P < 0.05$ and **, $P < 0.01$ vs corresponding WT controls (n = 6 biologically independent samples from different animals of independent experiment).

injection, ~0.11% and ~1.54% lymphocytes were EGFP⁺ in the liver and epididymal adipose tissue of WT recipient mice, respectively (Supplementary Fig. 3). Almost all EGFP⁺ cells were CD4⁺CD25⁺Foxp3⁺ with positive staining of neuropilin-1 (Nrp1), a specific marker for tTreg[30]. By contrast, there were significantly less amounts (~0.04% and ~0.74%, respectively) of EGFP⁺ cells present in the liver and epididymal adipose tissue of AKO recipient mice (Supplementary Fig. 3).

In TNC samples of WT mice adoptively transferred with EGFP⁺ thymocytes collected from the thymus of Adn-Cre/

ROSA^mT/mG mice, the adiponectin-expressing cells were in close contact with Nrp1 protein signals, but located away from the area with positive staining of galectin-3 (Fig. 4a), a glycoconjugate-binding protein involved in thymocyte-stromal cell interactions[31]. In TNC samples of AKO mice adoptively transferred with the EGFP⁺ thymocytes collected from the thymus of Adn-Cre/ROSA^mT/mG mice, however, the majority of adiponectin-expressing cells were not interacting with Nrp1 protein signals but in close contact with galectin-3 (Fig. 4b). Consistently, the distribution of adiponectin protein was

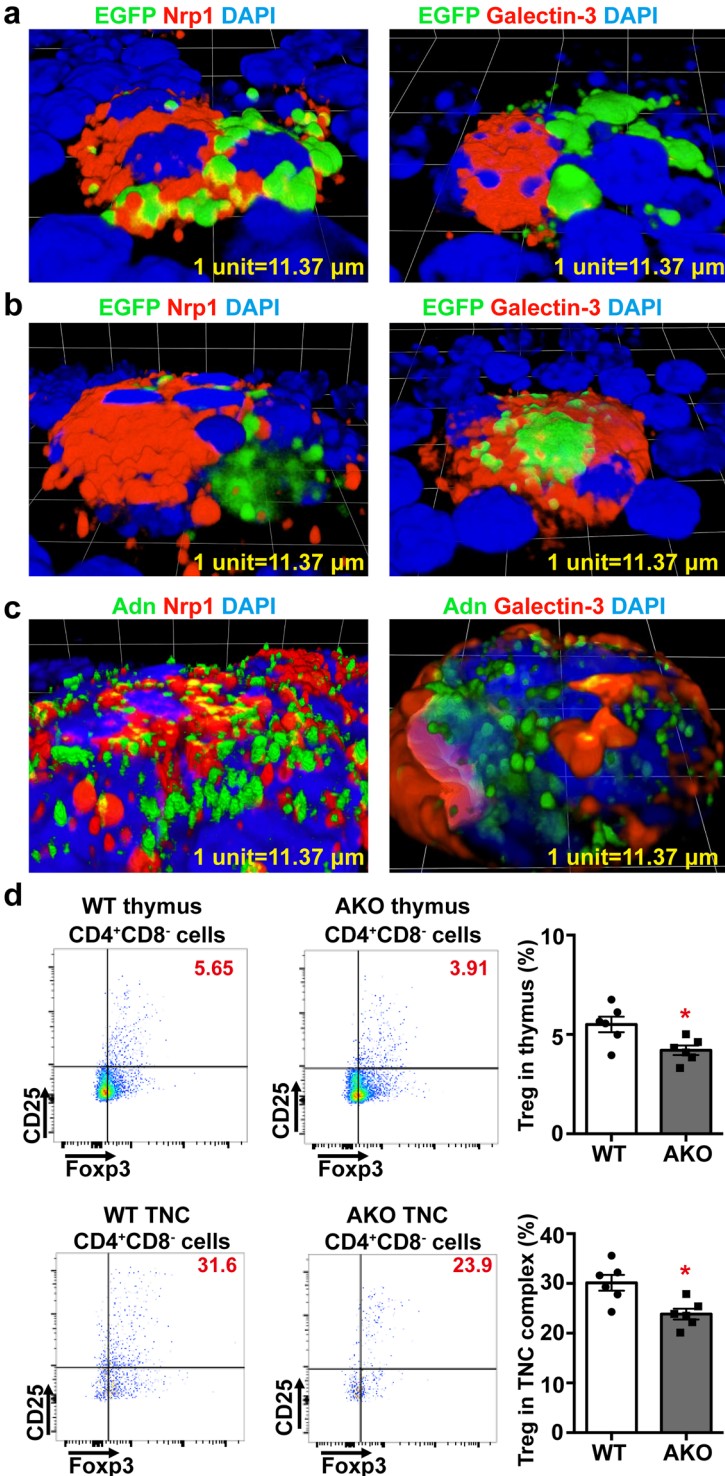

**Fig. 4 Mice lacking adiponectin show reduced number of tTreg in thymus and TNC complexes. a** Adiponectin-expressing EGFP+ cells isolated from Adn-Cre/ROSA^mT/mG were administered into 4-week-old WT mice [30000 cells/*mouse* via tail vein]. At 15-days after injection, TNC complexes were collected from thymus of the recipient mice for confocal microscopic analyses of the EGFP+ cells (green). Immunofluorescence counterstaining was performed to detect neuropillin-1 [Nrp1] or galectin-3 (red). **b** Adiponectin-expressing EGFP+ cells isolated from Adn-Cre/ROSA^mT/mG mice were administered into 4-week-old AKO mice [30000 cells/*mouse* via tail vein]. At 15-days after injection, TNC complexes were collected from thymus of the recipient mice for analyses as in (**a**). **c** TNC complexes were isolated from 6-week-old WT mice to detect the protein distribution of adiponectin [Adn], Nrp1 and galectin-3 by immunofluorescent staining and confocal microscopic analyses. **d** WT or AKO mice were sacrificed at the age of 7 weeks. Flow cytometry was performed to analyze the populations of CD4+CD8−CD25+Foxp+ tTreg in the thymus and TNC complexes for comparison. Data are shown as means ± SEM. *, $P < 0.05$ vs corresponding WT samples ($n = 6$ biologically independent samples from different animals of independent experiment).

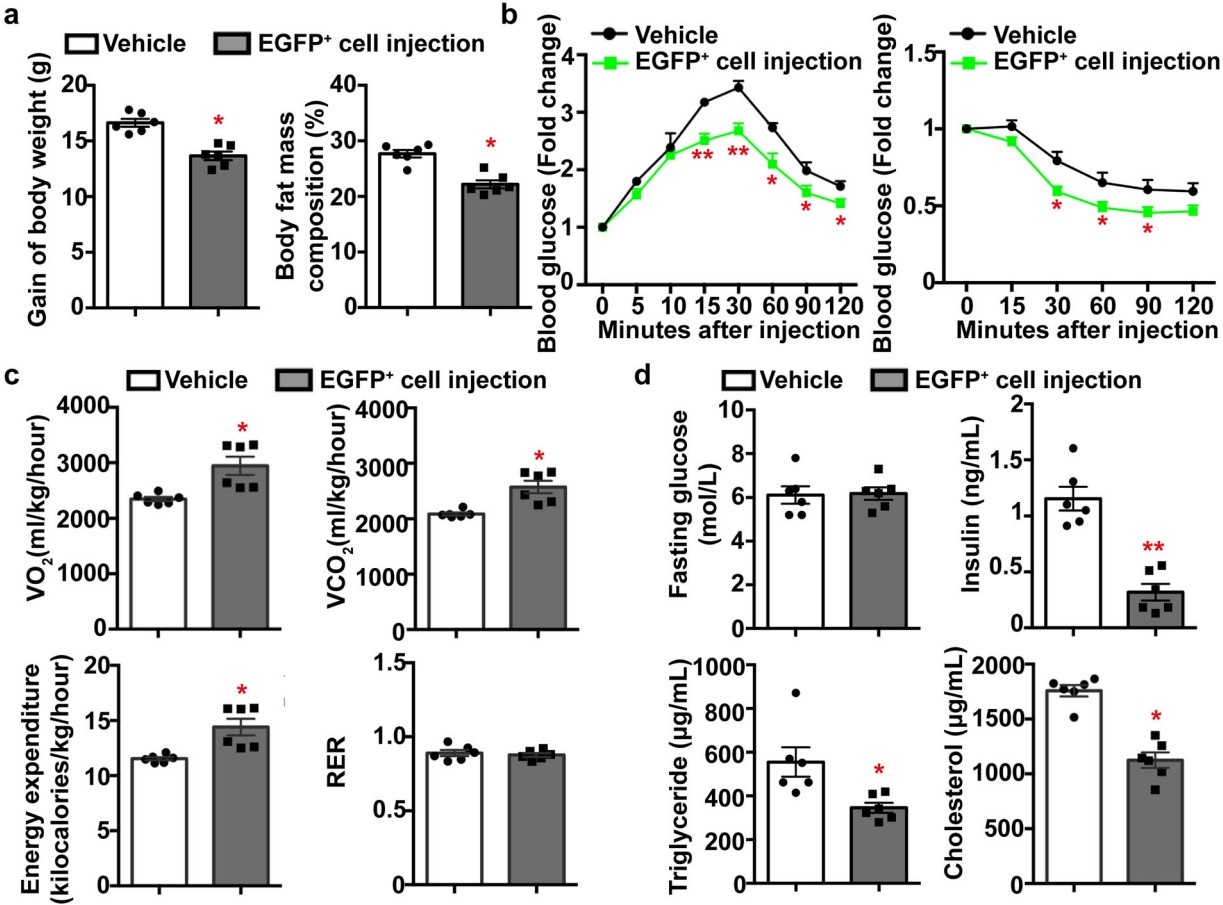

**Fig. 5 Treatment with adiponectin-expressing tTreg precursors alleviates HFD-induced metabolic abnormalities.** Vehicle or adiponectin-expressing EGFP$^+$ cells [30000 cells/*mouse*] isolated from Adn-Cre/ROSA$^{mT/mG}$ mice were injected via tail vein into 4-week-old WT mice, which were then subjected to HFD feeding for another 12 weeks. **a** At the end of treatment, the gain of body weight and the percentage body fat mass composition were calculated for comparison. **b** After 10- and 12 weeks of HFD, intraperitoneal glucose (*left*) and insulin (*right*) tolerance tests were performed as described in the Methods. Results are presented as fold changes against the glucose levels at time zero for comparison. **c** Indirect calorimetry was used to examine the VO$_2$, VCO$_2$, energy expenditure, and RER as described in the Methods. The 24-h average values were calculated for comparison. **d** The fasting blood glucose and serum insulin, triglyceride or cholesterol levels were measured after 12 weeks of HFD feeding for comparison. Data are presented as mean ± SEM. *, $P < 0.05$ and **, $P < 0.01$ vs corresponding vehicle control groups ($n = 6$ biologically independent samples from different animals of independent experiment).

intercalated with that of Nrp1 but at different regions from that of galectin-3 in TNC complexes of WT mice (Fig. 4c). Moreover, the total numbers of tTreg in the thymus and TNC complexes of AKO mice were significantly reduced when compared to those of WT animals (Fig. 4d).

The above results suggest that adiponectin-expressing cells are tTreg precursors, characterized by high expression of CD117 as well as CD4$^+$CD25$^+$CD8$^-$, and developed into mature tTreg primarily within the TNC complexes.

**Insulin-sensitizing activity of the adiponectin-expressing tTreg precursors.** Adoptive transfer of the adiponectin-expressing EGFP$^+$ cells was performed in 4-week-old male WT mice (30,000 cells/*mouse* by tail vein injection). After 12 weeks of high-fat diet (HFD) feeding, the gain of body weight and percentage body fat mass composition were significantly less in EGFP$^+$ cell-treated mice than those of vehicle controls (Fig. 5a). Mice treated with adiponectin-expressing EGFP$^+$ cells exhibited significantly improved glucose tolerance and insulin sensitivity (Fig. 5b), enhanced oxygen consumption (VO$_2$), CO$_2$ production (VCO$_2$) and energy expenditure, but similar respiratory exchange ratio (RER) when compared to those of the vehicle

control group (Fig. 5c). While there was no significant difference in fasting glucose levels, the plasma concentrations of insulin, triglyceride, and cholesterol were all significantly reduced in mice treated with adiponectin-expressing EGFP$^+$ cells (Fig. 5d). Moreover, the circulating liver injury markers, including alanine (ALT) and aspartate (AST) transaminases, were significantly decreased in mice treated with adiponectin-expressing EGFP$^+$ cells (185.01 ± 75.71 and 118.08 ± 42.87 U/L, respectively), when compared to the vehicle control group (300.28 ± 74.01 and 199.20 ± 69.64 U/L, respectively). Note that the same treatment with adiponectin-expressing EGFP$^+$ cells did not significantly improve the metabolic functions of AKO mice challenged with HFD (Supplementary Fig. 4).

In the blood circulation, mice treated with adiponectin-expressing EGFP$^+$ cells contained a significantly increased amount of CD3$^+$CD4$^+$ and CD3$^+$CD8$^+$ T-cells (Fig. 6a). In liver, the percentage number of Treg significantly increased, whereas that of Th17 cells significantly decreased in mice treated with adiponectin-expressing EGFP$^+$ cells (Fig. 6b). Note that the EGFP$^+$ cells characterized by CD4$^+$CD8$^-$CD25$^+$Foxp3$^+$ and positive Nrp1 staining were present in liver of mice treated with adiponectin-expressing tTreg precursors (Supplementary Fig. 5).

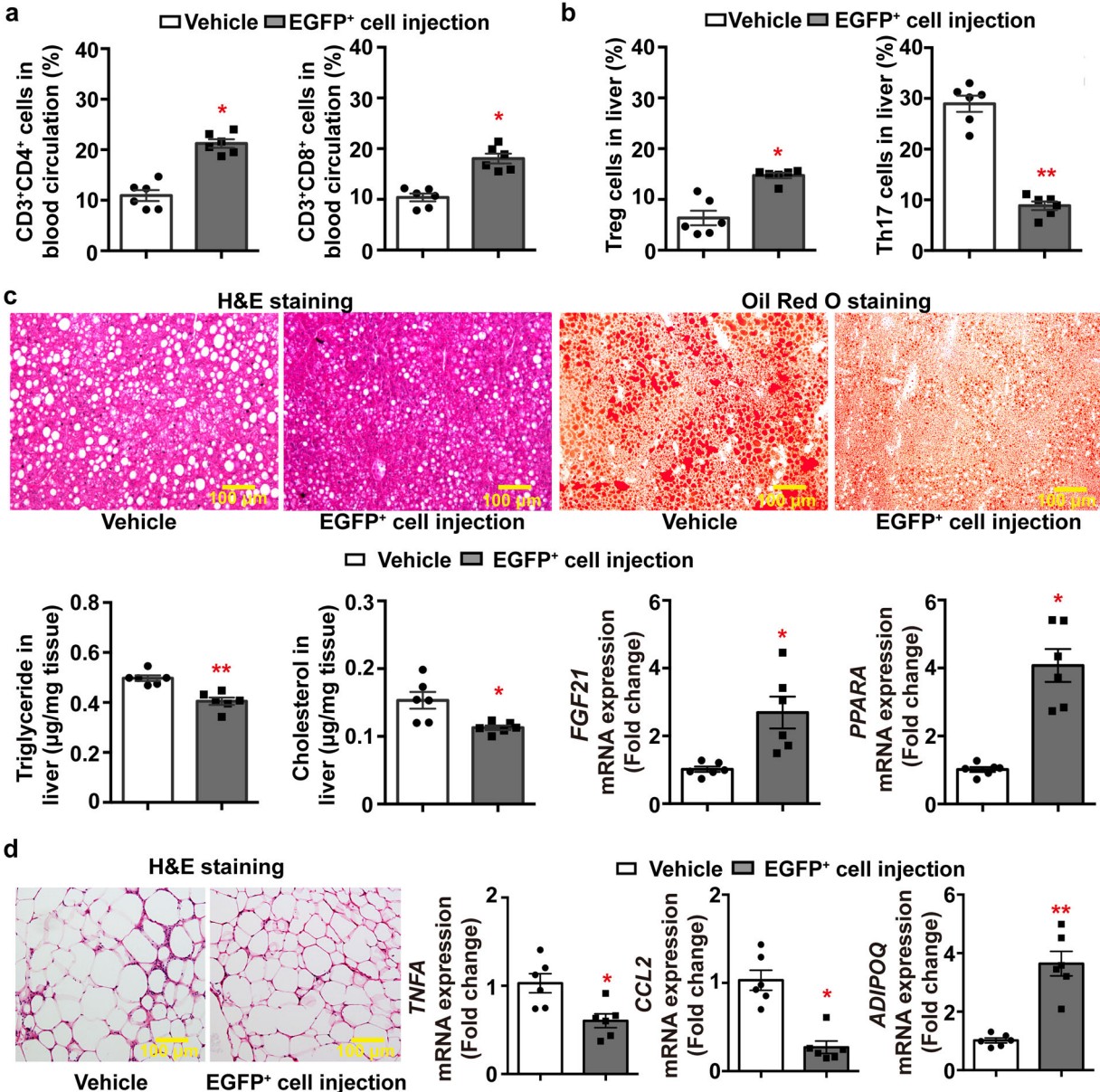

**Fig. 6 Treatment with adiponectin-expressing tTreg precursors alleviates HFD-induced tissue injuries.** Vehicle or adiponectin-expressing EGFP⁺ cells [30000 cells/*mouse*] isolated from Adn-Cre/ROSA^mT/mG mice were injected via tail vein into 4-week-old WT mice, which were then subjected to HFD feeding for another 12 weeks. At the end of treatment, blood, liver, and epididymal adipose tissue were collected for analyses. **a** Flow cytometry was performed to analyze the amount of CD3⁺CD4⁺ and CD3⁺CD8⁺ cells in the blood circulation for comparison. **b** Flow cytometry was performed to compare the amount of Treg and Th17 cells in liver tissues for comparison. **c** Hematoxylin and Eosin [H&E] or Oil Red O staining were performed for liver tissue sections to evaluate the accumulation and distribution of lipid droplets (*top*). The triglyceride and cholesterol contents in liver samples were examined by biochemical assays (*bottom left*). QPCR was performed for measuring the gene expression levels of *FGF21* and *PPARA* in liver (*bottom right*). **d** H&E staining was performed for epididymal adipose tissue sections (*left*). QPCR was performed for measuring the gene expression levels of *TNFA*, *CCL2* and *ADIPOQ* in epididymal adipose tissues (*right*). Data are presented as mean ± SEM; *, $P < 0.05$ and **, $P < 0.01$ vs corresponding vehicle controls ($n = 6$ biologically independent samples from different animals of independent experiment).

HFD-induced hepatic steatosis was significantly attenuated by the treatment with adiponectin-expressing tTreg precursors, as demonstrated by histological staining, tissue lipid measurement and mRNA quantification of genes encoding fibroblast growth factor 21 (*FGF21*) or peroxisome proliferator-activated receptor alpha (*PPARA*) (Fig. 6c). In epididymal adipose tissue, treatment with adiponectin-expressing tTreg precursors significantly reduced the average adipocyte size ($4751.6 \pm 442.0\ \mu m^2$ vs $5505.9 \pm 254.6\ \mu m^2$ in mice of vehicle controls, $P < 0.05$), the crown-like structures and

mRNA expression levels of inflammatory markers, including tumor necrosis factor alpha (*TNFA*) and monocyte chemoattractant protein-1 (*CCL2*), but significantly increased the *ADIPOQ* transcript levels (Fig. 6d).

Collectively, the data demonstrate that adiponectin-expressing tTreg precursors elicit potent insulin-sensitizing, hepatoprotective and anti-inflammatory activity via regulating T-cell homeostasis in the circulation and the immune microenvironment in peripheral organs.

**Anti-breast cancer activity of the adiponectin-expressing tTreg precursors**. The anti-tumor activity of adiponectin-expressing EGFP+ cells was evaluated in female MMTV-PyVT mice, which develop aggressive mammary tumors from the age of 7 or 8 weeks[13]. Adoptive transfer was performed in 4-week-old MMTV-PyVT animals (30,000 cells/mouse) by intravenous injection of EGFP+ thymocytes collected from the thymus of Adn-Cre/ROSA^mT/mG mice. Mammary tumor development was monitored every week until the age of 14 weeks. Compared to vehicle controls, treatment with adiponectin-expressing EGFP+ cells significantly inhibited the development of mammary tumor (Fig. 7a, left). The weights of tumor were reduced by over ~45% and those of the lung decreased by ~25% in MMTV-PyVT mice treated with adiponectin-expressing EGFP+ thymocytes (Fig. 7a, right).

Flow cytometry was performed to analyze the composition of T lymphocyte subsets. In blood samples collected from 14-week-old MMTV-PyVT mice, the total amounts of CD3+CD4+ and CD3+CD8+ were both significantly increased by treatment with the adiponectin-expressing EGFP+ thymocytes (Fig. 7b). The percentage contents of CD4+ and CD8+ cells were also significantly increased in mammary tumors of MMTV-PyVT mice treated with adiponectin-expressing EGFP+ thymocytes (Supplementary Fig. 6a). A distinct population of Nrp1+ Treg cells, characterized by CD4+CD25^highFoxp3+, was present in mammary tumors of MMTV-PyVT mice treated with adiponectin-expressing EGFP+ cells (Fig. 7c). Compared to the CD4+CD8−CD25^low populations, the CD4+CD8−CD25^high cells were less effective in suppressing the proliferation of CD8+ cytotoxic T lymphocytes (Supplementary Fig. 6b). Compared to those of the vehicle group, the mRNA expression levels of transforming growth factor beta (*TGFB1*), *CCL2*, interleukin-6 (*IL6*) and vascular endothelial growth factor alpha (*VEGFA*) were significantly downregulated in tumor samples of MMTV-PyVT mice treated with adiponectin-expressing EGFP+ cells (Fig. 7d).

Compared to MMTV-PyVT mice containing wild-type *ADIPOQ* alleles (PyVT-WT), those lacking adiponectin expression (PyVT-AKO) were not responsive to the same treatment with EGFP+ thymocytes collected from the thymus of Adn-Cre/ROSA^mT/mG mice (Supplementary Fig. 7). In PyVT-AKO mice, the CD3+CD4+ T-cells were significantly reduced, accompanied by an augmented number of immature CD3−CD4+CD8+ cells in the blood circulation (Supplementary Fig. 8a). The latter population of cells promoted mammary tumor development in NOD/SCID mice implanted with human breast cancer MDA-MB-231 cells (Supplementary Fig. 8b). In blood samples of MMTV-PyVT mice treated with adiponectin-expressing EGFP+ cells, the amount of CD3−CD4+CD8+ thymocytes were significantly decreased when compared to vehicle controls (Supplementary Fig. 8c).

In summary, the results suggest that adiponectin-expressing tTreg precursors exert anti-breast cancer activity at least partly via modulating the repertoire of T-cells in the blood circulation as well as mammary tumor tissues.

**Adiponectin facilitates T-cell selection within the TNC complexes**. Flow cytometry was performed to evaluate the CD3+ thymocytes in the thymus and enriched TNC samples collected from 7-week-old WT and AKO mice. Compared to those of WT animals, the percentage contents of CD3+ cells were significantly reduced in both thymus and TNC complexes of AKO mice (Fig. 8a). In the thymus of WT mice treated with adiponectin-expressing tTreg precursors, the amount of CD3+ and CD4+ SP cells significantly increased when compared to the vehicle control animals (Fig. 8b). The number of TNC complexes, characterized

by CD326+β5t+PI^high [32], was significantly decreased in the thymus of AKO mice when compared to WT animals (Fig. 8c). Moreover, the two-color staining of CD45 for extracellular (eCD45) and intracellular (iCD45) thymocytes revealed that the percentage amount of thymocytes attached to TNC complexes was significantly less in preparations from AKO mice than those of WT samples (Fig. 8c). The TNC complexes of AKO mice were smaller and enclosed a significantly reduced number of CD3+ thymocytes (Fig. 8d). Compared to those of AKO mice, the TNC complexes of WT showed an increased expression level and a distinct distribution pattern of the thymus-specific proteasome subunit β5t, i.e. the finger-like projections surrounding the released lymphocytes (Fig. 8d).

Thymus tissues were collected from WT and AKO mice to examine the microenvironment by immunofluorescence staining of the epithelial markers. While there were no significant changes in the distribution of p63 and galectin-3, the β5t signals showed a clear boundary at the cortico-medullar junction in AKO thymus (Supplementary Fig. 9). Compared to WT thymus, the number of cells co-stained with antibodies against β5t and cytokeratin 5 was significantly decreased (Supplementary Fig. 9). Live cell imaging was performed with TNC complexes isolated from Adn-Cre/ROSA^mT/mG mice lacking the *ADIPOQ* alleles (Adn-Cre/ROSA^mT/mG-AKO). There were few cells labeled with mT fluorescence released from the TNC complexes, which did contain the EGFP+ cells (Supplementary Movie 2). With time, the number of apoptotic cells within the TNC complexes, as demonstrated by positive staining for the diamidino-2-phenylindole (DAPI) DNA-specific dye (Supplementary Movie 2), was significantly less than those isolated from Adn-Cre/ROSA^mT/mG mice (Supplementary Movie 1).

The results conjointly indicate that adiponectin produced by thymocytes within the TNC complexes plays an important role in the selection, development, and maturation of T lymphocytes in thymus.

**Adiponectin regulates the expression and distribution of CD100**. CD100, also known as semaphorin 4D, is a leukocyte cell surface glycoprotein and the first semaphorin member characterized in the immune system[33]. The protein was detected in the TNC complexes of WT and AKO mice, however, with significantly different patterns of distribution (Fig. 9a). In TNC complexes isolated from WT thymus, CD100 exhibited distinct close associations with its high-affinity receptor plexin B1 (Fig. 9a, top left). The interactions with CD72, a low-affinity receptor of CD100, were also detectable at the periphery region of TNC complexes of WT mice (Fig. 9a, top right). In TNC complexes isolated from AKO thymus, CD100 was widely distributed around plexin B1 and presented as a diffusible semaphorin filling in the extracellular space between the thymocytes and the epithelial plasma membrane (Fig. 9a, bottom left). There were hardly any co-localization signals between CD100 and CD72 within the TNC complexes of AKO mice (Fig. 9a, bottom right).

Using the polyclonal antibody recognizing the fragment between amino acid 812 and 862, a 150-kDa CD100 was detected in both thymus and TNC complexes isolated from the WT and AKO mice (Fig. 9a, left). Compared to the thymus samples, the relative amount of 150-kDa CD100 was significantly increased in TNC complexes. Using the polyclonal antibody recognizing the fragment between amino acid 502 and 636, a 120-kDa CD100 was detected and significantly decreased in the TNC complexes isolated from WT mice (Fig. 9b, left). The relative ratio between 150-kDa and 120-kDa CD100 was significantly higher in WT TNC complexes than those of AKO samples (Fig. 9b, right). Immunoprecipitation was performed using the above two

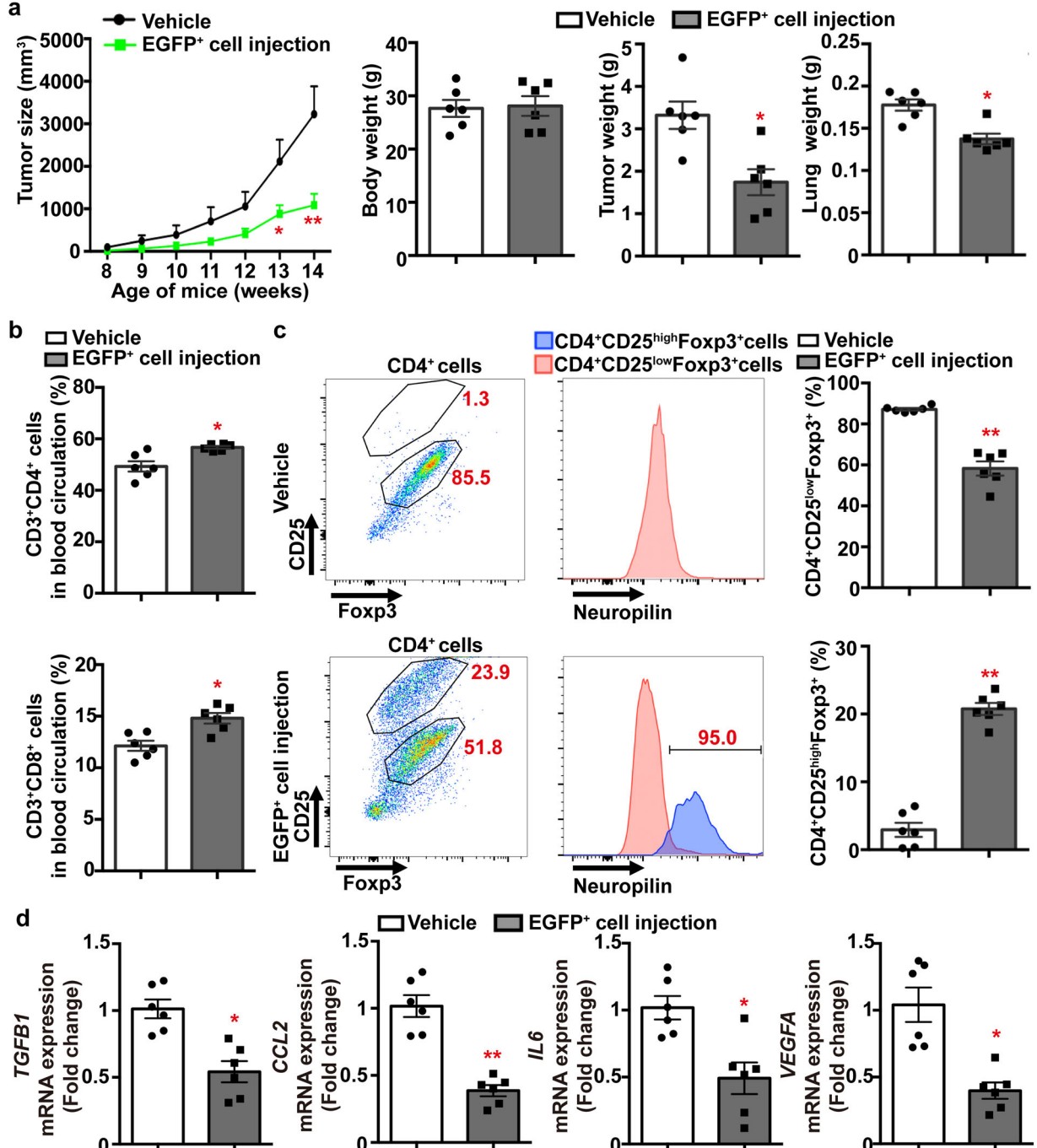

**Fig. 7 Treatment with adiponectin-expressing thymocytes inhibits breast cancer development in MMTV-PyVT mice.** Vehicle or EGFP+ cells collected from the thymus of 4–5-week-old Adn-Cre/ROSA^mt/mg mice were adoptively transferred into 4-week-old MMTV-PyVT mice via tail vein injection [30,000 cells/*mouse*]. Mammary tumor development was monitored once per week as described in the Methods. At the age of 14 weeks, mice were sacrificed to collect blood, tumor, and lung samples for analyses. **a** The tumor growth, body weights, tumor and lung tissue weights were measured and calculated for comparison. **b** Flow cytometry was performed to analyze the percentage composition of CD3+CD4+ and CD3+CD8+ T-cells in blood samples. **c** Flow cytometry was performed to analyze the Treg subpopulations in mammary tumors based on the antibody staining of CD4, CD25, Foxp3 and Nrp1. The percentage composition of CD4+CD25^lowFoxp3+ and CD4+CD25^highFoxp3+ Treg were calculated for comparison. **d** The mRNA expression levels of genes including *TGFB1*, *CCL2*, *IL6* and *VEGFA* were measured in tumor samples by QPCR. Data are presented as mean ± SEM. *, $P < 0.05$ and **, $P < 0.01$ vs corresponding vehicle controls ($n = 6$ biologically independent samples from different animals of independent experiment).

polyclonal antibodies recognizing different regions of CD100 (Fig. 9c). The results demonstrated that adiponectin bound to CD100 (Fig. 9c, *top left*), consistent with the immunofluorescence co-staining results in TNC complexes isolated from WT mice (Fig. 9c, *top right*). By contrast, the protein-protein interactions between CD100 and galectin-3 were only detectable in TNC

samples isolated from AKO, but not in the preparations of WT mice (Fig. 9c, *bottom left*). The extensive co-localization signals between CD100 and galectin-3 were further confirmed by immunofluorescence staining in TNC complexes of AKO mice (Fig. 9c, *bottom right*). The phosphorylation levels of Lyn, a predominant Src-family kinase regulating self-tolerance[34], were

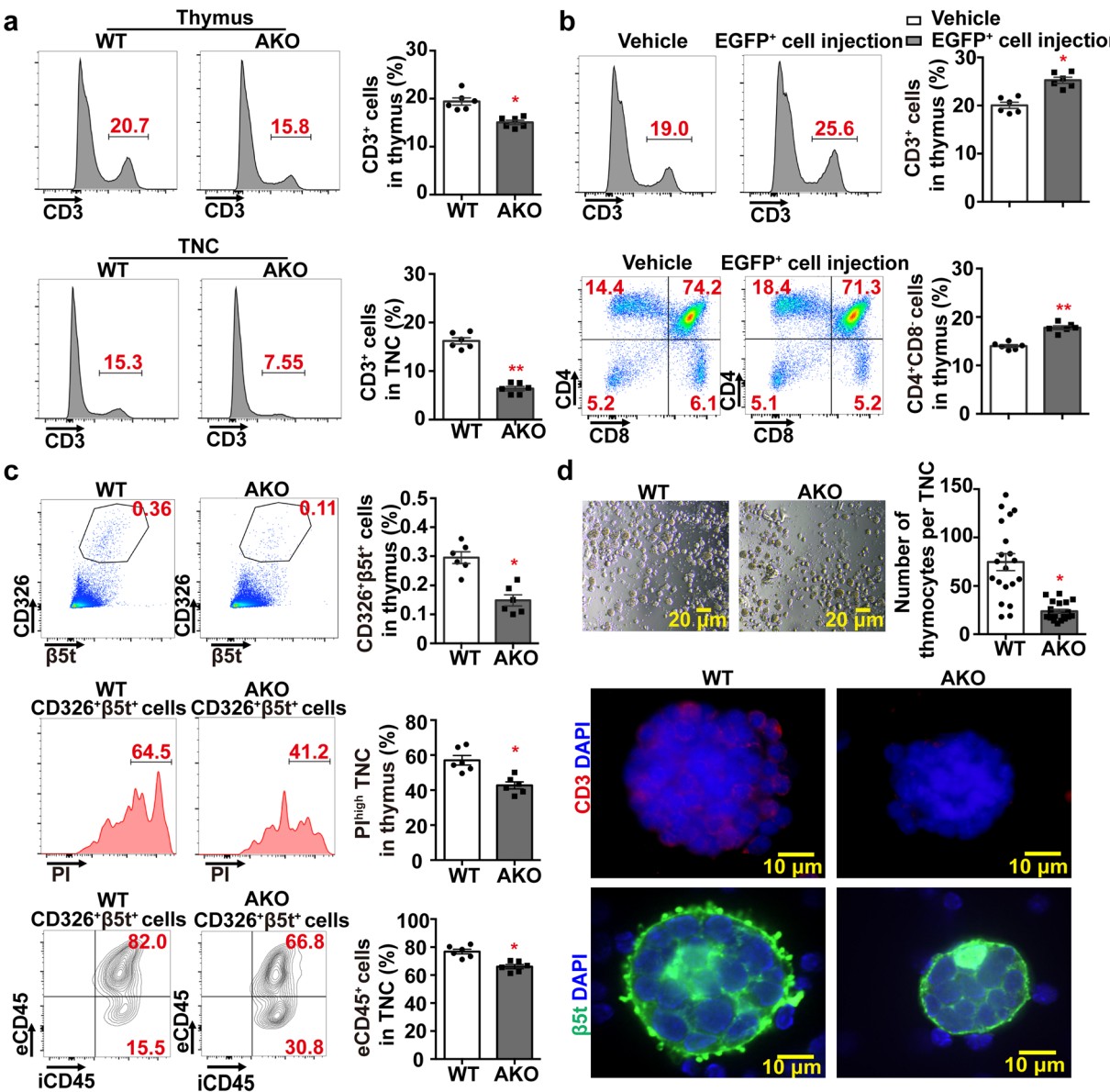

**Fig. 8 Adiponectin facilitates T-cell selection and development within the TNC complexes. a** WT and AKO mice were sacrificed at the age of 6–7 weeks. Thymus (*top*) and TNC complexes (*bottom*) were prepared to analyze the percentage of CD3+ cell population by flow cytometry. **b** Four-week-old WT mice were subjected to vehicle treatment or adoptive transfer with EGFP+ cells from thymus of Adn-Cre/ROSAmT/mG mice (30000 cells/*mouse* via tail vein) as described in the Methods. At 21 days after the injection, thymus was collected and subjected to flow cytometry analyses for measuring the percentage contents of CD3+ (*top*) or CD4+CD8− (*bottom*) cells. **c** WT and AKO mice were sacrificed as in (**a**) to prepare thymic cell suspensions. Flow cytometry was performed to analyze the total amount of CD326+β5t+ TNC complexes (*top*), TNC complexes with high expression levels of propidium iodide [PI] (*middle*), or eCD45+ cells attached outside of the TNC complexes (*bottom*). **d** WT and AKO mice were sacrificed as in (**a**) to prepare the thymic cell suspensions enriched with TNC complexes. Inverted microscope was used to visualize and manually count the number of thymocytes within each TNC complex (*top*). Immunofluorescence staining was performed to detect CD3 (*middle*) and β5t (*bottom*) signals in TNC complexes of WT and AKO mice. The slides were counterstained with DAPI. Data are presented as mean ± SEM. *, *P* < 0.05 and **, *P* < 0.01 vs corresponding controls (*n* = 6 biologically independent samples from different animals of independent experiment).

significantly increased by 2.5-fold in AKO thymus (Supplementary Fig. 10).

Taken together, the results indicate that by modulating CD100 expression and distribution, adiponectin is involved in the process of antigen presentation within the TNC complexes.

## Discussion

Treg maintains the immune homeostasis by facilitating self-tolerance, in turn preventing autoimmune as well as chronic inflammatory diseases[35]. Treg-based therapy is considered as a promising approach to treat metabolic disorders, including obesity, insulin resistance and type 2 diabetes[36–38]. However, the optimal application of Treg relies on further understanding of the origin, specificity, and function of their different subsets. Results of the present study demonstrate that adiponectin is expressed in the thymus, by a subpopulation of Treg precursors. Adiponectin-expressing thymocytes are of hematopoietic origin, differentiate and mature into tTreg within the lymphoepithelial TNC complexes, a niche microenvironment for the establishment of central

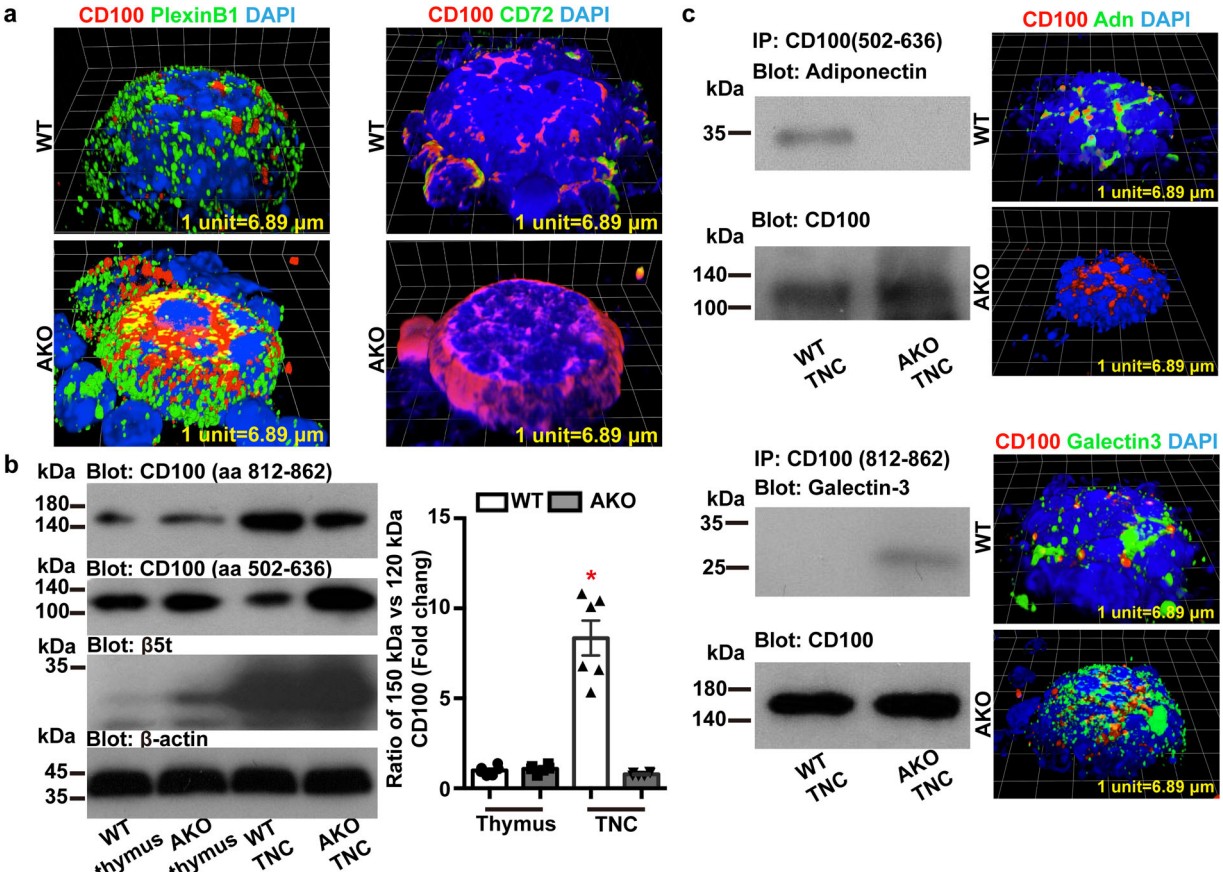

**Fig. 9 Adiponectin regulates the expression and distribution of CD100 within the TNC complexes.** WT or AKO mice were sacrificed at the age of 7 weeks to collect the thymus for subsequent analyses. **a** TNC complexes were prepared for immunofluorescence staining with antibodies against CD100 [Cat No. 565556 from BD Biosciences (*left*) or Cat No. Ab231961 from Abcam (*right*)], plexin B1 or CD72. Confocal microscopic analyses revealed different patterns of CD100 protein distribution in relation to plexin B1 or CD72 in the TNC complexes from WT and AKO mice, respectively. **b** Western blotting was performed to analyze the protein expression of CD100 in thymus and TNC lysates prepared from WT and AKO mice, using antibodies recognizing different regions [from amino acid 812–862 or 502–636] of this molecule. Beta-actin [β-actin] was detected as the loading controls and β5t probed as a marker for TNC complexes. **c** TNC complexes were prepared for co-immunoprecipitation using antibodies recognizing CD100 from amino acid 502–636 (*top*) or amino acid 812–862 (*bottom*). The presence of adiponectin or galectin-3 in the immune complexes was confirmed by Western blotting using specific antibodies (*left*). Immunofluorescence staining and confocal microscopic analyses were performed to analyze the interactions between CD100 and adiponectin or galectin-3 within the TNC complexes (*right*). Data were presented as mean ± SEM. *, $P < 0.05$ vs WT thymus ($n = 6$ biologically independent samples from different animals of independent experiment).

tolerance[39,40]. After tail vein injection, these cells rapidly resided within the TNC complexes and differentiated into tTreg. Adoptive transfer of adiponectin-expressing tTreg precursors not only effectively prevented HFD-induced obesity, insulin resistance and nonalcoholic fatty liver injuries in WT mice, but also inhibited the breast cancer development in MMTV-PyVT mice. Within the TNC complexes, adiponectin plays an important role in the development of T lymphocyte and the formation of tTreg, thus promoting the immunological self-tolerance and the systemic immune-metabolic homeostasis.

The classical route of tTreg formation involves first the induction of CD25, which confers high-affinity IL-2 binding and subsequent expression of Foxp3[41]. A large portion of Treg are produced in the thymus from CD25hiCD4+CD8− precursors. An alternate route characterized by the reciprocal induction of Foxp3 and CD25 is also suggested[42]. In this regard, CD25 is required to rescue the developing tTreg cells from Foxp3-induced apoptosis. No matter tTreg differentiation proceeds via CD25+Foxp3− or CD25−Foxp3+ intermediates, both pathways involve apoptosis-prone transitional stages at which cells compete for cytokines as survival factors. However, the precise signals that divert

thymocytes into tTreg remain largely undefined. The present results demonstrate that adiponectin-expressing tTreg are likely to differentiate from the CD117+CD4+CD25+ precursors via the CD4+CD8+ DP stage in the thymus. After adoptive transfer, about 28 and 8% of adiponectin-expressing thymocytes differentiate into mature tTreg in the WT and AKO thymus, respectively, despite that the precursors exhibited similar capacities to enter the TNC complexes. At 15-days after adoptive transfer, the total number of adiponectin-expressing tTreg in AKO thymus is less than 10% of those in WT thymus. By contrast, the percentage composition of EGFP+CD4+CD8+ DP cells is significantly higher in AKO thymus than that of WT thymus, indicating that adiponectin is involved in the formation and maturation of tTreg from DP cells. Mice without adiponectin show a significantly reduction of tTreg in both thymus and TNC complexes.

It remains unclear how the lineage restriction of tTreg is determined in thymus. The selection of tTreg is driven by T-cell receptor (TCR) specificity and the efficiency directly correlated with negative selection[26]. Thus, tTreg development is accompanied by clonal deletion of the conventional T-cells. The fate determination between clonal deletion and diversion to the tTreg

is influenced by intrathymic self-antigen expression and presentation—ubiquitous antigen expression leads to clonal deletion, whereas tissue-restricted antigen expression promotes tTreg lineage diversion[43]. Currently, there is no evidence suggesting that a single type of thymic antigen-presenting cell (APC) contributes to tTreg differentiation in thymus. Apart from dendritic and epithelial cells, B-cells act as APC to present endogenous self-antigen for tTreg formation and the establishment of T-cell tolerance[44]. The available evidence suggests a role of TNC complexes in the development of tTreg and the establishment of central tolerance. The number of TNCs is significantly decreased in the thymic cortices of autoimmune mice[40]. The mRNA transcript and protein of Foxp3 are present in TNC complexes, so do the tissue-restricted antigens[45]. Functional loss of TNC complexes is involved in various autoimmune diseases. The locally produced adiponectin is widely distributed between the engulfed thymocytes and the specialized membrane folds of TNC complexes. In particular, the presence of adiponectin influences the expression and distribution of CD100, a surface glycoprotein of the semaphorin family involved in antigen presentation and T/B-cell interactions[33]. Adiponectin binds directly to CD100 and inhibits the cleavage of the 150-kDa full-length molecule to form the 120-kDa species that interacts with plexin B1, a high-affinity receptor on epithelial cells[46], or other surface glycoproteins, such as galectin-3. In WT TNC complexes, CD100 presents mainly as the 150-kDa species to form the membrane tethering structures containing adiponectin, Nrp1 and/or CD72, a low-affinity receptor expressed on B-cells[47]. When binding to CD72, CD100 suppresses the negative signals to enhance the response of B-cell antigen receptor[48]. Thus, adiponectin may facilitate the CD100/CD72-mediated T/B interactions within the TNC complexes. By recognizing the self-antigens presented by B-cells, adiponectin-expressing tTreg develop in TNC complexes of WT thymus and distribute in peripheral organs to promote the immunological self-tolerance and immune-metabolic homeostasis. By contrast, the ubiquitous antigen presentation via CD100/plexin B1 high-affinity interactions triggers excessive clonal deletion thus leading to a reduced number of CD3$^+$ thymocytes in AKO thymus.

The majority of circulating Treg are generated in thymus, with only a small amount of peripheral Treg induced from CD4$^+$CD25$^-$ naïve T-cells. In early life, the newly generated tTreg cells populate peripheral lymphoid and non-lymphoid organs in the body[49]. Treg resided in peripheral organs are highly heterogeneous, consisting of different subsets with distinct specificity and functional properties[37,49]. In response to the specific cues in the environment, tTreg alter their functional properties by switching on the expression of different molecular signatures specific to T helper (Th) lineages, including Th1, Th2, and Th17 cells[50,51]. Foxp3-dependent repression of effector cytokine gene expression, including IL-4, IFN-γ, and IL-17, is essential for the maintenance of Treg homeostasis[52]. The loss-of-Treg stability leads to immune-mediated diseases[35]. Impairment of Nrp1 facilitates the reprograming of Treg[53]. By contrast, semaphorin-Nrp1 signaling axis acts to maintain Treg stability[54]. Under inflammatory conditions, Treg develop an IL-17-secreting capability, which reduces their suppressive activity. HFD leads to a decrease of Treg and an increase of Th17 cells in the liver, causing the progression of nonalcoholic fatty liver disease[38]. Here, the results demonstrate that in WT mice, treatment with adiponectin-expressing tTreg precursors significantly attenuates HFD-induced metabolic abnormalities, including systemic insulin resistance, fatty liver injuries, and adipose tissue inflammation. Treatment with adiponectin-expressing thymocytes leads to an increase of Treg and a decrease of Th17 cells in the liver, which at least partly contribute to the prevention of HFD-induced hepatic

steatosis and injury in WT mice. It is possible that the expression and presence of adiponectin within TNC complexes regulate the stability of tTreg by modulating the semaphorin-Nrp1 signaling axis via the surface expression of CD100. The composition of Treg in visceral adipose tissue has long-term and profound effects on systemic insulin sensitivity and metabolic function[55]. By controlling the chronic mild inflammation in adipose tissue, Treg play a protective role in obesity and insulin resistance[49,56–58]. Note that mice treated with adiponectin-expressing tTreg exhibit a significantly enhanced *ADIPOQ* expression in adipose tissue, which also contributes to the improved systemic insulin sensitivity and reduced fatty liver injuries. It would be interesting to dissect the precise interrelationships between adipocyte- and thymus-derived adiponectin, especially in the context of Treg regulation.

TNC complexes not only provide a microenvironment for the establishment of T-cell tolerance, but also provide a special microenvironment for T-cell differentiation and selection[32]. They consist of large epithelial cells in the cortex and cortico-medullary junction of the thymus. In TNC complexes, the extensions of plasma membrane form a cage-like structure to internalize thymocytes and facilitate membrane interlocking for the presentation of major histocompatibility complex (MHC) class I and II antigens to TCR[40]. Thymocytes that interact with TNC complexes include the αβTCR$^{lo}$CD4$^+$CD8$^+$ DP cells[32]. DP thymocytes bearing TCR to recognize peptide-MHC complexes with low avidity are positively selected and differentiated into SP cells. Only 3–5% of developing thymocytes survive this checkpoint during T-cell development. The vast majority of DP thymocytes undergo negative selection and clonal deletion, which relies on strong affinity of TCR for MHC self-peptide complexes. The negatively selected thymocytes are trafficked into the cytoplasm of the TNC complexes and degraded by lysosomes[59]. Compared to those in WT samples, the number of CD3$^+$ thymocytes is significantly decreased in thymus and TNC complexes of AKO mice. Despite the information, the intrinsic mechanism by which adiponectin regulates the fate of CD4$^+$CD8$^+$ DP cells within the TNCs complex, i.e. to either die through apoptosis, undergo differentiation into tTreg, or mature to the αβTCR$^{high}$CD69$^+$CD4$^+$CD8$^+$ stage for further development, remain unknown. Without adiponectin, the selection function of TNC complexes is impaired, causing the escape of immature CD4$^+$CD8$^+$ thymocytes from thymus and their release into the blood circulation to facilitate tumor development in PyVT-AKO mice. In TNC complexes of AKO mice, the enhanced galectin-3/CD100 interactions may inhibit lymphoepithelial interactions and facilitate the release of premature DP thymocytes from the TNC complexes[31]. Treatment with adiponectin-expressing tTreg precursors facilitates lymphoepithelial interactions as well as the selection and development of T lymphocytes in TNC complexes, thus preventing the release of immature CD4$^+$CD8$^+$ thymocytes in the blood circulation. Note that in primary tumors of breast cancer patients, the CD4$^+$CD8$^+$ cells are significantly increased[60]. In tumors of PyVT-WT mice treated with adiponectin-expressing thymocytes, two populations of Treg are detected with either CD25$^{hi}$ or CD25$^{low}$ and Nrp1$^+$ or Nrp1$^-$ expressions, which exhibit different capacities to suppress CD8$^+$ cell proliferation. However, whether they represent subpopulations of Treg that elicit pro- or anti-tumor activity warrants further investigation[61–66]. Taken together, adiponectin-expressing tTreg precursors elicit anti-cancer activity at least partly by facilitating T-cell selection in TNC complexes, suppressing the release of immature CD4$^+$CD8$^+$ cells from thymus, and promoting the immune cell recruitment within the tumor microenvironment.

In summary, the results of the present study collectively suggest that adiponectin-expressing tTreg exert a profound immuno-modulating, insulin-sensitizing, hepatoprotective and anti-tumorigenic activity. Apart from the cell-autonomous role in tTreg maturation, adiponectin promotes the selection of T lymphocytes and facilitates the production of $CD3^+$ T-cells within the TNC complexes of thymus. However, whether or not it represents a unifying mechanism explaining the pleiotropic roles of adiponectin in various pathophysiological conditions warrants further investigations.

## Methods

**Mouse models**. All mice were housed in a room under the controlled temperature ($23 \pm 1\,^\circ\text{C}$) and 12-h light-dark cycles, with free access to water and standard mouse chow (4.07 kcal/g; LabDiet 5053; LabDiet, Purina Mills, Richmond, VA, U.S. A.). All experimental procedures were approved by the Committee on the Use of Live Animals in Teaching and Research, the University of Hong Kong and carried out in compliance with the Guide for the Care and Use of Laboratory Animals published by the National Institutes of Health (8th Edition, 2011; https://www.ncbi.nlm.nih.gov/books/NBK54050/).

Mice with (WT) or without (AKO) the *ADIPOQ* alleles were maintained on both C57BL/6 J and FVB/N background. For metabolic evaluations, WT and AKO of C57BL/6 J background were fed with high-fat diet (19.33 kcal/g from 49.85% fat, 20% protein, and 30.15% carbohydrate; D12451; Research Diet, New Brunswick, NJ, U.S.A.) to induce dietary obesity[67,68]. For studies related to mammary tumor development, FVB/N-Tg (MMTV-PyVT)634 Mul/J [002374 from Jackson Laboratory (Bar Harbor, ME, U.S.A.)] were crossbred with AKO of FVB/N background to produce mice with (PyVT-WT) or without (PyVT-AKO) the *ADIPOQ* alleles[13,14]. ROSA$^{mT/mG}$ (007676; Jackson Laboratory) mice were crossbred with Tg(Adipoq-cre)1Evdr/J (010803; Jackson Laboratory) to obtain the transgenic two-color reporter mouse model, referred to as Adn-Cre/ROSA$^{mT/Mg}$. The Adn-Cre/ROSA$^{mT/mG}$ mice were crossbred with the *ADIPOQ* conditional knockout (Adn-CKO) mice with both exon 2 and 3 deleted to produce Adn-Cre/ROSA$^{mT/mG}$-AKO mice lacking the expression of adiponectin. The genotyping primers for the above mouse models are listed in Supplementary Table 1.

**Adoptive cell transfer**. Cells expressing EGFP fluorescence (EGFP$^+$) were freshly sorted from the thymus of Adn-Cre/ROSA$^{mT/mG}$ mice. Around $3 \times 10^4$ EGFP$^+$ cells were injected via tail vein into 4–5-week-old WT and AKO, or PyVT-WT and PyVT-AKO mice. The WT and AKO mice were subsequently fed with high-fat diet for another 12 weeks, during which their body weight, fat mass composition, and insulin sensitivities were monitored on a weekly basis[69]. Tumor development in PyVT-WT and PyVT-AKO mice were monitored on a weekly basis[2,13]. For evaluating the EGFP$^+$ cell identities and development, the recipient mice were given 500 rad (5 Gy) irradiation for 4–6 h before the adoptive transfer.

**Orthotopic inoculation of human breast cancer cells**. Nonobese diabetic/severe combined immunodeficient NOD.CB17-Prkdc$^{scid}$/J (NOD/SCID; 001303) mice were obtained from the Jackson Laboratory. Mice were anesthetized with a mixture of ketamine (90 mg/kg ip), xylazine (20 mg/ml ip) and acepromazine (1.8 mg/kg ip). Human breast cancer MDA-MB-231 cells ($2 \times 10^5$), together without or with the $CD4^+CD8^+$ cells ($3 \times 10^3$), were inoculated into the right thoracic mammary fat pads. The development of mammary tumors was monitored twice per week with a vernier caliper and calculated using the formula [sagittal dimension (mm) × cross dimension (mm)$^2$] /2.

**Flow cytometry and fluorescence-activated cell sorting (FACS)**. Multicolor flow cytometry and cell sorting were performed with BD LSR Fortessa Analyzer (BD Bioscience, San Jose, CA, U.S.A.) and BD FACSAria$^{TM}$ SORP Cell Sorter (BD Bioscience), respectively. The cytometer performance was checked prior to each experiment. The fluorescent lights activated by a solid-state laser of 405 nm (FVD eFluor 506, VioBlue), 488 nm (FITC, PE, PerCP, PE-Vio770) and 640 nm (APC, APC-Vio770) were collected by photomultiplier tubes, voltage of which was adjusted using the cell sample and fixed for all experiments. The fluorescence compensation was performed using a single-antibody-labeled COMPtrol Goat anti-Mouse Ig (H&L) Particle Kit (Spherotech, Lake Forest, IL, U.S.A.) and then with the single-antibody-labeled cell samples. For each sample, at least 10,000 events were acquired for analyses. Data analyses were performed using Diva 6.1 and CellQuest (BD Biosciences) and FlowJo 10.0 software (TreeStar Inc., Ashland, OR, USA). Dead events were excluded by FSC-A/SSC-A gating and adhesion events excluded by FSC-A/FSC-H gating.

**Preparation of cell suspensions**. Single cell suspension was prepared from thymus as described[70]. In brief, freshly collected tissues were cut, minced and transferred to Dulbecco's Modified Eagle Medium (DMEM) containing 2 mg/ml collagenase Type I (Gibico$^{TM}$, Waltham, MA, U.S.A.) and 40 µg/ml DNase I

(Sigma-Aldrich, St. Louis, MO, U.S.A.). After incubation at 37 °C with shaking for 30 min, cells were strained through a 100 µm filter mesh and centrifuged at $400 \times g$. The pellets were resuspended in phosphate-buffered saline (PBS) and then labeled with specific antibodies for subsequent flow cytometric analyses. Where indicated, TNC complexes were enriched from enzyme-digested thymic cell suspensions by four-step $1 \times g$ sedimentation in fetal bovine serum (FBS)[71]. Single cell suspension was obtained from the enriched TNC samples by gentle mechanical dissociation of the complexes with a 3 ml syringe and 29 G needle.

For peripheral blood analysis, EDTA was used as an anticoagulant and added at a concentration of 1.5 mg per ml of blood. The erythrocyte-lysing buffer (555899; BD Biosciences) was used to prepare blood sample for flow cytometric analysis and cell sorting. The mixed panel of BV421-conjugated anti-CD3, PE-CF594-conjugated anti-CD4 and Alexa Fluor 647-conjugated anti-CD8 was used to examine or sort T-helper, T-cytotoxic and immature $CD4^+CD8^+$ cells.

After perfusion with PBS through portal vein, liver tissues were dissected, homogenized and digested in DMEM containing 0.5 mg/ml collagenase Type IV (Gibico$^{TM}$) and 150 µg/ml DNase I (Sigma-Aldrich) at 37 °C for 40 min with shaking. After centrifugation at 1600 rpm for 5 min, cell pellets were resuspended in PBS and passed through a 100 µm cell strainer before loading onto a gradient containing 40 and 70% Percoll (GE Healthcare Bio-Sciences, Sweden). The fractionation was performed by centrifugation at $1126 \times g$ for 20 min at 4 °C. The middle layer containing lymphocytes was collected for subsequent analyses.

Epididymal adipose tissues were dissected, homogenized and digested in DMEM containing 1 mg/ml collagenase Type II (Gibico$^{TM}$) and 150 µg/ml DNase I (Sigma-Aldrich) at 37 °C for 30 min with shaking. After centrifugation at 1600 rpm for 10 min, cell pellets were resuspended in PBS and passed through a 100 µm cell strainer.

**Antibody labeling**. Antibodies were obtained from Biolegend (San Diego, CA, USA), BD Bioscience, eBioscience (San Diego, CA, USA) or Vector (Burlingame, CA, USA). For all staining, Fc receptors were blocked with anti-CD16/32 (eBioscience 14–0161–82) before antibody labeling. After centrifugation at $400 \times g$, cells were incubated with specific combinations of antibodies on ice for 30 min. The mixed panel of Pacific Blue$^{TM}$-conjugated anti-CD3 (Biolegend 100214), phycoerythrin (PE)-CF594-conjugated anti-CD4 (BD 562285), Alexa Fluor® 647-conjugated anti-CD8a (Biolegend 100724), fluorescein isothiocyanate (FITC)-conjugated anti-CD44 (BD 553133) and PE-Cy$^{TM}$7-conjugated anti-CD25 (BD 552880) was used for analyzing $CD4^-CD8^-$ double-negative (DN), $CD4^+CD8^+$ double-positive (DP), $CD3^+CD4^+$ or $CD3^+CD8^+$ single-positive (SP) cells. The mixed panel of BD Horizon™ V450-conjugated anti-lineage cocktail (BD 561301) including 500A2 recognizing mouse CD3e, M1/70 recognizing CD11b; RA3–6B2 recognizing CD45R/B220, TER-119 recognizing Ly-76 mouse erythroid cells, and RB6–8C5 recognizing Ly-6G and Ly-6C was used for cell depletion. FITC-conjugated anti-CD44, PE-Cy7-conjugated anti-CD25, allophycocyanin (APC)-conjugated anti-c-Kit (Biolegend 135108) and PE-conjugated anti-CD24 (BD 553262) was used to examine DN1–4 populations and early thymic progenitors (ETPs). The mixed panel of PE-conjugated anti-CD45 (Biolegend 103106), APC-conjugated anti-EpCAM (Biolegend 118214), PE-Cy7-conjugated anti-Ly51 (Biolegend 108314) and FITC-conjugated anti-UEA-I (Vector FL-1061) was used to evaluate the cortical (cTEC) and medullary (mTEC) thymic epithelial cells. For analyzing samples from Adn-Cre/ROSA$^{mT/mG}$ or Adn-Cre/ROSA$^{mT/mG}$-AKO mice, Pacific Blue$^{TM}$-conjugated anti-CD45 (Biolegend 103125), anti-CD4 (Biolegend 100427), anti-CD25 (Biolegend 102021) were used to avoid the interference from cells with green or red fluorescence. Fixed and permeabilized lymphocytes from liver were stained with PE-conjugated anti-IL17A (Biolegend 506903), BV421-conjugated anti-CD45 (Biolegend 103134) and APC-conjugated anti-CD4 (Biolegend 100412) to detect Th17 cells.

**Enrichment and labeling of TNC complexes**. Freshly collected thymus tissues were cut, minced and transferred to DMEM containing 2 mg/ml collagenase Type I and 40 µg/ml DNase I. After incubation at 37 °C with shaking for 30 min, cells were strained through a 100 µm filter mesh and centrifuged at $400 \times g$. The pellets were washed and resuspended in PBS for flow cytometric analyses. Briefly, the thymic cell suspensions were stained with antibodies including BV421-conjugated anti-CD45 (eCD45), APC-conjugated anti-EpCAM (CD326), anti-β5t antibodies (MBL Life science PD021) and Alexa Fluor 488-conjugated anti-rabbit IgG antibody (Abcam ab150077) in the dark on ice for 30 min, then fixed and permeabilized with Cytofix/CytopermTM Fixation/Permeabilization Kit (BD 554714). The cells were subsequently incubated with antibodies including PE-conjugated anti-CD45 (iCD45) and in the dark on ice for another 30 min. Where indicated, cells were stained with 10 µg/mL PI after fixation and permeabilization prior to the analyses.

**Confocal live cell imaging**. Enriched TNC samples from Adn-Cre/ROSA$^{mT/mG}$ or Adn-Cre/ROSA$^{mT/mG}$-AKO mice were cultured in phenol red free DMEM using a temperature-controlled (37 °C) chamber. Five percent $CO_2$ was continuously supplied during the experiment. The 3D live cell images were acquired every 15 min using the UltraVIEW® VOX Spinning Disc confocal system (Perkin Elmer) equipped with a confocal fluorescent microscopy (Inverted: Nikon Eclipse Ti-E) and a mecury/xenon/LED's lamp. DAPI (20 µg/ml; D9542, Sigma-Aldrich) was

added into the chamber before image acquisition, The Volocity® visualization and quantification software (Perkin Elmer) was used for image analyses. Both Max intensity and Iso surface mode were used to display the dynamic video.

**In situ hybridization**. Thymic cell suspensions were plated onto the slides using a Cytopro™ centrifuge (Wescor, Utah, USA). In situ hybridization was performed using the RNAscope® Intro Pack 2.5 HD Reagent Kit BROWN-Mm (Cat No 322371; Advanced Cell Diagnostics, Beijing, China), which contained eleven pairs of double "Z" oligo probes (RNAscope® Probe-Mm-Adipoq; Cat No 440051) to target the 1–640 bp of murine *ADIPOQ* (NCBI Reference Sequence: NM_009605.5). The hybridization signals were developed by applying the RNA-scope® DAB reagent (Advanced Cell Diagnostics). All sections were counterstained with hematoxylin before mounting on glass coverslips with a xylene based mounting medium.

**Reverse transcription (RT-PCR) and quantitative PCR (QPCR)**. RNAiso Plus (9109, TaKaRa, Japan) was used to isolate the total RNA from tissue or cell samples. After checking the quality by 2100 bioanalyzer (Agilent, Santa Clara, CA, USA), reverse transcription was performed using PrimeScript™ RT reagent Kit (RR037A, TaKaRa). RT-PCR was performed for amplifying the *ADIPOQ* transcript and the products analyzed by agarose gel electrophoresis. QPCR was performed using SYBR® Green reagents from Qiagen (Hilden, Germany). The reactions were carried out in a 7000 Sequence Detection System (Applied Biosystems, Foster City, CA, USA). Quantification was achieved by comparing the Ct values that were normalized with 18S rRNA or β-actin as the internal controls. The QPCR primers are listed in Supplementary Table 2.

**Histological analyses**. Tissues were cut into small pieces and fixed in 10% formalin solution for 48 h before transferring to 75% ethanol for long-term storage at 4 °C. The paraffin blocks were prepared for sectioning at 5 μm thickness. The tissue slides were stained with hematoxylin and eosin (H&E) solution. Lesions of liver were evaluated with NASH CRN Scoring System. The grading for steatosis included 0: <5%; 1: 5–33%; 2: 34–66%; and 3: >66%. For lobular inflammation, the grades were 0: none; 1: <2 foci/20× field; 2: 2–4 foci/20× field; 3: >4 foci/20× field. The scores for hepatocellular ballooning were 0: none; 1: mild, few; 2: moderate-marked, many. The NASH CRN scores ranging from 0 to 8 were calculated as the sum of steatosis score (0–3), lobular inflammation score (0–3), and hepatocellular ballooning score (0–2)[72]. After H&E staining of the adipose tissue sections, the size of adipocytes was measured and calculated using Image J software (Version 1.51, NIH, USA). Ten fields were randomly chosen and the size of adipocytes is presented as average cross-sectional area. Frozen liver tissues were embedded in Tissue-Tek OCT compound (Sakura® Finetek, CA, U.S.A.), sectioned at 5 μm and then stained with Oil Red O (Sigma-Aldrich) for 10 min. All slides were examined under Olympus biological microscope BX41, and images were captured using an Olympus DP72 color digital camera.

**Immunofluorescence staining**. Thymic cell suspensions were plated onto the slides using a Cytopro™ centrifuge (ELITech, U.K.). Cells were fixed with 4% paraformaldehyde on ice for 10 min or cold acetone at −20 °C for 5 min, blocked with 5% bovine serum albumin (BSA) for 30 min, and then incubated with specific antibodies (Supplementary Table 3) in a humidified chamber overnight at 4 °C. After washing, various fluorophore-conjugated secondary antibodies were applied to the sample slides for 1-h incubation at room temperature in the dark. Thymus tissues were fixed with 10% formalin, embedded in paraffin, and cut as five μm micron sections. The tissue sections were deparaffinized, rehydrated, and boiled in sodium citrate buffer (10 mM, 0.05% Tween 20, pH 6) for 10 min. Endogenous peroxidase was inactivated by incubation of slides with 3% $H_2O_2$ for 15 min. After blocking with 5% BSA at room temperature for 1 h, the tissue sections were incubated with primary antibodies (Supplementary Table 3) in a humidified chamber overnight at 4 °C. The protein targets were visualized by incubating with fluorophore-conjugated secondary antibodies at room temperature in the dark. Images were captured under the fluorescence microscope (Leica Microsystems, Bensheim, Germany) and analyzed using the AxioVision Imaging Plus software (Carl Zeiss Ltd., United Kingdom).

**Western blotting and ELISA**. Total tissue or cell lysates were separated by SDS-PAGE, transferred to polyvinylidene difluoride membranes, and then probed with various antibodies (Supplementary Table 3). After incubation with secondary antibodies, the antibody-antigen complexes were detected using an enhanced chemiluminescence kit from GE Healthcare (RPN2209; Uppsala, Sweden). Murine adiponectin immunoassay kit (32010; Immunodiagnostics, Hong Kong, China) was used for quantifying adiponectin levels in serum samples and tissue extracts collected from different mouse models. Briefly, 100 μl of the diluted serum samples (1:600) or tissue extracts (40 μg total protein) were used for the measurement by following the manufacturer's instructions. Absorbance at 450 nm was recorded to determine the presence and quantity of murine adiponectin[73].

**Evaluation of metabolic function**. Body weight and fat mass composition were measured between 1000 and 1200 h for mice that were either starved overnight or fed ad libitum. The body mass composition was assessed in conscious and unanesthetized mice using a Bruker minispec Body Composition Analyzer (Bruker Optics, Inc., Woodlands, TX). Blood glucose was monitored by tail nicking using an Accu-Check Advantage II Glucometer (Roche Diagnostics, Mannheim, Germany). The intraperitoneal glucose tolerance test (ipGTT) and insulin tolerance test (ITT) were performed using mice that were fasted overnight and for 6 h, respectively, as described[69]. In brief, for ipGTT, mice were given a glucose load by intraperitoneal injection (1 g of glucose/kg of body weight). For ITT, mice were intraperitoneally injected with insulin (1 unit/kg of body weight). Plasma glucose levels were measured at different time points as indicated. Circulating and tissue contents of lipids, including triglycerides, total cholesterols, were analyzed using LiquiColor Triglycerides and Stanbio Cholesterol (Stanbio Laboratory, Boerne, TX) and the Half-Micro Test Kit (Roche Diagnostics), respectively. The fasting serum insulin concentration was quantified using the commercial ELISA kits from Mercodia AB (Uppsala, Sweden). Metabolic rate (VO2, VCO2, and respiratory exchange ratio [RER]) was measured by indirect calorimetry using a six-chamber open-circuit Oxymax system component of the Comprehensive Laboratory Animal Monitoring System (CLAMS; Columbus Instruments, Columbus, OH). All mice were acclimatized to the cage for 48 h before recording the parameters.

**Statistics and reproducibility**. All experiments were performed with six to eight samples per group and results derived from at least three independent measurements. Values are expressed as mean ± SEM. The statistical analyses were performed using the Statistical Package for the Social Sciences version 11.5 software package (SPSS, Inc., Chicago, IL, USA). Comparison between groups was performed using Student's unpaired *t*-test or two-way ANOVA (GraphPad Prism 7.00 Software, Inc., San Diego, CA, USA). *P* values less than 0.05 were accepted to indicate statistically significant differences.

**Reporting summary**. Further information on research design is available in the Nature Research Reporting Summary linked to this article.

## Data availability

The authors declare that all data generated or analyzed during the study are included in this published article and its supplementary information files. In addition, the original datasets are deposited and available at https://doi.org/10.25442/hku.13503291.

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

## Acknowledgements

This work was supported by the grants from Seeding Funds for Basic Research of the University of Hong Kong, the General Research Funds (17124420 and 17153016) and Collaborative Research Funds (C7037-17W and C7017-18G) of Research Grant Council, the Areas of Excellence Scheme (AoE/M-707/18) of University Grants Committee.

## Author contributions

Y.Z., H.C., J.C., and Y.L. performed the experiment; Y.Z., H.C., J.C., and Y.W. analyzed the data; Y.Z., H.C., and Y.W. wrote the manuscript; AX and YW designed the project and supervised the study.

## Competing interests

The authors declare no competing interests.
