## [Peer Review File · Communications Biology]

Reviewers' comments:

Reviewer #1 (Remarks to the Author):

In this study, Zhang et al. reported the identification of adiponectin in a population of regulatory T-cells (Treg) in thymus. In addition, they showed that the expression of adiponectin in the thymic nurse cell (TNC) complexes of thymocytes facilitated the maturation of T lymphocytes in the thymus via modulating CD100 expression and distribution. By using the transgenic two-color reporter mouse model (Adn-Cre/ROSA^{mT/mG}) and adiponectin-knockout (AKO) mice, the authors showed that adoptive transfer of adiponectin-expressing thymocytes not only alleviated HFD-induced metabolic abnormalities and but also inhibited breast cancer development. This study has identified a novel and intriguing role of adiponectin-expression thymocytes in the establishment of central tolerance and the findings are interesting and potentially significant. However, there are a number of important questions need to be addressed before the work is suitable for publication in *Communication Biology*.

Critiques:

1. What is the signaling mechanism by which adiponectin facilitates the maturation of tTreg? Are adiponectin receptors and other signaling components highly expressed in Treg cells?
2. What are the mechanisms by which adiponectin-expressing tTreg cells improve HFD-induced obesity and enhance insulin sensitivity?
3. Could the expression of Cre recombinase driven by the Adipoq promoter be detected in other tissues or cells of the Adn-Cre/ROSA^{mT/mG} such as bone marrow and hemopoietic stem cells as the authors mentioned in the Introduction?
4. The adiponectin expressed in the thymus were mainly in the form of hexamer (Fig.1A). Is the expression of this form of adiponectin correlated with its specific function in the thymocytes? Are adiponectin expression and oligomerization in the thymus altered under pathophysiological conditions such as obesity?
5. In figure 3, the authors found that on day 15 after injection, the CD4⁺CD8⁺ double positive T cells were increased but the mature thymic Treg cells and the total number of EGFP⁺ cells were significantly decreased in recipient AKO mice compared with recipient WT mice (Fig.3C-3D). Since the donor cells were the same. it is unclear whether the different microenvironment of recipient WT and AKO mice affected the donor cell development. It is also unclear why the adiponectin-expressing donor cells could not support their own development in the AKO recipient mice?
6. T cells from AKO mice had less Treg cells in the peripheral, which was prone to induce EAE diseases (Laura Piccio et al, *European journal of immunology*,2013). It would be interesting to know whether the development of Treg cells was affected in the thymus of AKO mice.
7. The authors showed that compared to vehicle controls, treatment with adiponectin-expressing EGFP⁺ cells significantly inhibited the development of mammary tumors (Fig. 6A). However, this treatment also markedly increased the proportion of CD4⁺CD25^{high}Foxp3⁺Nrp1^{high}Treg cells in the tumors (Fig. 6C). Since Treg cells can promote immune tolerance and accelerate tumor growth, it is unclear how treatment with adiponectin-expressing EGFP⁺ cells suppress tumor development.
8. Most information would be helpful to better understand the data shown in Fig. 8. For example, what is the percentage of tumor-killing IFN- γ ⁺ Th1 and CD8 T cells in the tumor? Is there a functional difference between the two isoforms of CD100? Does the interaction between CD100 and adiponectin or galectin-3 play distinct roles in regulating Treg cell biology and function? How adiponectin regulates the expression of distribution of CD100?
9. The authors concluded that adiponectin plays a role in establishing central tolerance by

modulating CD100 expression and distribution. However, the data were correlative and it is unclear whether adiponectin plays a distinct role in CD100 KO thymus. Thus, a less strong tone of the conclusion may be more appropriate.

Reviewer #2 (Remarks to the Author):

In this study, the authors demonstrated that adiponectin-expressing cells exist in thymus and these cells are differentiated into Treg cells within thymic nurse complexes (TNCs). Adoptive transfer of the adiponectin-expressing cells improved HFD-induced glucose intolerance and liver steatosis. Also, these cells suppressed tumor growth in MMTV-PyVT mouse by inhibiting egression of premature CD3-CD4+CD8+ cells from thymus. Furthermore, they claimed that adiponectin-expressing thymocytes regulate expression and distribution of CD100. These results seem to be interesting and novel. However, it appears that there are several points to be resolved. Particularly, the underlying mechanisms by which adiponectin-expressing thymocytes regulate above biological events are not fully elucidated. In addition, the quality of several data is not sufficient to determine whether the data were correctly analyzed. Followings are specific issues to improve the study:

1. The authors used EGFP+ cells as adiponectin-expressing cells in adiponectin-cre/ROSA^{mT/mG} mouse model. However, due to irreversible changes mediated by cre-loxP system, it is elusive whether EGFP+ cells consistently express adiponectin or not. To address this issue, adiponectin expression level should be tested in EGFP+ thymocytes compared to EGFP- ones. Also, adiponectin protein level in the thymus of EGFP+ cell-injected AKO mice should be compared with that of WT counterparts. These experiments would be the key prerequisite step to understand the novel functions of adiponectin-expressing thymocytes.
2. In Figure 4 and 5, administration of EGFP+ cells showed dramatic improvement of metabolic parameters in adipose tissue and liver. However, it is still unknown whether these effects would be mediated by T cell development in thymus or directly occurred in adipose tissue and liver. To clarify this issue, it is necessary to show how many EGFP+ thymocytes were infiltrated into above organs and to examine the relative ratio of EGFP+ cells among resident-Treg cells. Also, it should be investigated the degree of changes in the amount of adiponectin in above tissues (e.g., thymus, adipose tissue, and liver) before and after EGFP+ cell injection. These experiments could help to explain the mechanism by which EGFP+ thymocytes regulate whole body metabolism.
3. The authors claimed that most of EGFP+ cells were colocalized with TNCs in thymus. However, there is no direct evidence supporting this statement because they only showed confocal images of single TNC, but not that of other regions such as medullar regions. In Figure 1B, adiponectin signal seems to be found in medullar regions. Thus, it is required to provide additional tissue section images that show EGFP signal and TNC markers (e.g., CK5, CK8) in overall thymus, including TNC and other regions. Moreover, quantitation data of colocalization signals would be helpful to support their model/proposal.
4. The current study includes several western blot data. However, some of them need to be improved. For example, in Figure 1A, adiponectin band was cropped. Also, in Figure 8B, there are no positive markers that verify the credibility of TNC fractionation. It is needed to provide full blot images of western blots and conduct additional western blotting to show successful fractionation of TNC.

Re: Submission of the manuscript entitled “**Adiponectin-expressing Treg facilitate T lymphocyte development in thymic nurse cell complexes**” (COMMSBIO-20-1888A)

The detailed point-by-point responses are listed in the following sections:

1) Data confirming whether EGFP⁺ cells in the thymus are still expressing adiponectin.

Answer: The EGFP⁺ and EGFP⁻ cells were sorted from the thymus of Adn-Cre/ROSA^{mT/mG} mice for analyses. Both *ADIPOQ* transcript and adiponectin protein were detected in EGFP⁺ thymocytes (**Supplementary Figure 2, C and D**). Detailed descriptions are included in **the 2nd paragraph on page 6** of the revised manuscript. Please also refer to the responses to Q15 of Reviewer #2 for more explanations.

2) Data showing whether the effect of EGFP⁺ thymocyte transfer originates in the thymus or in the adipose tissue and liver themselves.

Answer: After adoptive transfer, the distribution of EGFP⁺ cells were analysed in different tissues of the recipient mice. On the 1st and 3rd day after injection, EGFP⁺ cells were not detectable in liver and adipose tissue of the recipient mice. On the 15th day after injection, EGFP⁺ cells were present in liver and adipose tissue as mature Treg (CD4⁺CD8⁻CD25⁺Foxp3⁺) with positive staining of surface marker Nrp1⁺ (**Supplementary Figure 3**). The data support that thymus development is essential for EGFP⁺ precursors to become mature tTreg before moving to peripheral organs. Detailed descriptions are included in **the 3rd paragraph on page 6** of the revised manuscript.

3) Stronger data showing that EGFP⁺ cells are preferentially co-localised with thymic nurse cells. This could be achieved by showing images of the whole tissue.

Answer: We thank the reviewers’ suggestions. In the revised **Figure 1B**, the co-staining immunofluorescence images with antibodies against adiponectin and cytokeratin 5 or 8 were included. In the revised **Figure 2C**, the fluorescence images for both thymocyte cell suspensions and enriched TNC complexes from the thymus of Adn-Cre/ROSA^{mT/mG} mice were included. Detailed descriptions are included in **the 3rd paragraph on page 4** and **the 2nd paragraph on page 6** of the revised manuscript. Please also refer to the responses to Q17 of Reviewer #2 for more explanations.

4) A more solid rationale for why the transfer of EGFP⁺ cells results in better anti-tumor responses.

Answer: Thymus plays an important role in the antitumor immunity. Immunosenescence in thymus results in reduced output of naïve T-cells and a restricted T cell receptor (TCR) repertoire, thus dampening the immune surveillance of neoplasia (Wang E et al Front Immunol 2020 Apr 30;11:773. doi: 10.3389/fimmu.2020.00773). In tumor tissues, more and more evidence suggesting that there exists diversified populations of Treg (Yano H et al Immunology 2019 Jul;157(3):232-247. doi: 10.1111/imm.13067). Moreover, Treg ablation enhances mammary tumor development and metastasis (Martinez LM et al Front Immunol. 2019 Aug 29;10:1942. doi: 10.3389/fimmu.2019.01942). Thus, the role of Treg in tumor development needs to be dissected in a context-dependent manner. In the present study, after adoptive transfer, the EGFP⁺ precursors enter thymus and reside into TNC complexes to facilitate the selection and development of T-cells including Treg, which not only enhances the T-cell repertoire but also

reduces the production of immature T-cells (**Figure 6, Figure 7 and Supplementary Figure 7**). The EGFP⁺ Treg are present in tumor tissues together with other tTreg characterized by Nrpl⁺ (**Figure 7**). The different subpopulations of Treg elicit distinct functions on the suppression of cytotoxic CD8⁺ T-cells (**Supplementary Figure 6**), and the regulation of inflammatory tumor microenvironment (**Figure 6**). Additional information is also added in the discussion to explain the anti-tumor activity of adiponectin-expressing tTreg precursors (**the 1st paragraph on page 11** of the revised manuscript).

5) Discussion and/or extra data defining why the EGFP⁺ cells differentiate into Treg.

Answer: We thank the editor/reviewers' suggestions. Accordingly, detailed information is provided about the routes of tTreg formation in thymus (**the 2nd paragraph on page 16**), the close relationship between tTreg development and clonal deletion (**the 3rd paragraph on page 17**), the thymic niche microenvironment for tTreg development and the establishment of central tolerance (**the 3rd paragraph on page 17**) and the possible role of adiponectin in enhancing self-antigen presentation by B-cells within the TNC complexes (**the 3rd paragraph on page 17**) in the revised discussion. Additional data is also included in **Figure 2D** to demonstrate the differentiation of EGFP⁺ cells into Treg with detailed descriptions in **the 2nd paragraph on page 6** of the revised manuscript.

Reviewer #1

6) What is the signaling mechanism by which adiponectin facilitates the maturation of tTreg? Are adiponectin receptors and other signaling components highly expressed in Treg cells?

Answer: At this stage, the precise signaling mechanisms by which adiponectin facilitates the maturation of tTreg are not clear. However, our results in **Figure 7** and **Figure 8** suggest that adiponectin produced locally within the TNC complexes is involved in the lymphoepithelial interactions via regulating CD100 shedding. By binding to and preventing the shedding of CD100 located at the surface of lymphocytes, adiponectin facilitates the weak T/B interactions mediated by CD72 and thus the antigen presentation for tTreg development. By contrast, adiponectin deficiency enhances the ubiquitous antigen presentation via CD100/plexin B1 high affinity interactions, which trigger excessive clonal deletion thus leading to a reduced number of CD3⁺ thymocytes in AKO thymus. The relevant information has been added in **the 2nd paragraph of discussion**.

7) What are the mechanisms by which adiponectin-expressing tTreg cells improve HFD-induced obesity and enhance insulin sensitivity?

Answer: The results in **Figure 4** and **Figure 5** demonstrated that treatment with adiponectin-expressing tTreg precursors significantly increases Treg and decrease Th17, i.e. the Treg stability, in liver of WT mice challenged with HFD. In both liver and adipose tissue, reduced inflammation contributes to the enhanced insulin sensitivity and energy metabolism. The relevant discussion has been added in **the 2nd paragraph on page 9** of the revised manuscript.

8) Could the expression of Cre recombinase driven by the Adipoq promoter be detected in other tissues or cells of the Adn-Cre/ROSA^{mT/mG} such as bone marrow and hemopoietic stem cells as the authors mentioned in the Introduction (which step express adn, expression time)?

Answer: In the revised manuscript, new data of *ADIPOQ* expression in different subpopulations of thymocytes have been included (**Supplementary Figure 1**). The results demonstrate that *ADIPOQ* is expressed in early thymic progenitors of the T-lineage, as well as CD4⁺ SP and CD4⁺CD8⁺ DP cells. However, we prefer not to divert the efforts for studying hematopoietic stem cells or other tissues at this stage, but focusing on the central regulation of Treg in thymus.

9) The adiponectin expressed in the thymus were mainly in the form of hexamer (Fig.1A). Is the expression of this form of adiponectin correlated with its specific function in the thymocytes? Are adiponectin expression and oligomerization in the thymus altered under pathophysiological conditions such as obesity?

Answer: We thank the reviewer's comments. So far, our results do not suggest any significant differences in the formation of adiponectin oligomers in thymus from those in adipose tissues.

10) In figure 3, the authors found that on day 15 after injection, the CD4+CD8+ double positive T cells were increased but the mature thymic Treg cells and the total number of EGFP+ cells were significantly decreased in recipient AKO mice compared with recipient WT mice (Fig.3C-3D). Since the donor cells were the same. it is unclear whether the different microenvironment of recipient WT and AKO mice affected the donor cell development. It is also unclear why the adiponectin-expressing donor cells could not support their own development in the AKO recipient mice?

Answer: We appreciate the reviewer's careful observations and comments. Yes, the development of EGFP⁺ is defective in AKO thymus. Both the total number and the percentage Treg are significantly less in AKO thymus, when compared to the WT recipient (**Figure 3**). Note that in TNC complexes of AKO recipient mice, the adiponectin protein produced by the EGFP⁺ cells is very limited when compared to those in the WT TNC complexes (**Figure 3A**). Thus, in terms of adiponectin levels represent one of the major differences in the TNC microenvironment of WT and AKO recipient mice.

11) T cells from AKO mice had less Treg cells in the peripheral, which was prone to induce EAE diseases (Laura Piccio et al, European journal of immunology,2013). It would be interesting to know whether the development of Treg cells was affected in the thymus of AKO mice.

Answer: Although the data were not shown, we have consistently observed a decreased amount of Treg in both thymus and TNC complexes of AKO mice when compared to those of WT mice. The information has been added in the **2nd paragraph on page 12** of the revised manuscript.

12) The authors showed that compared to vehicle controls, treatment with adiponectin-expressing EGFP+ cells significantly inhibited the development of mammary tumors (Fig. 6A). However, this treatment also markedly increased the proportion of CD4+CD25^{high}Foxp3⁺Nrp1^{high}Treg cells in the tumors (Fig. 6C). Since Treg cells can promote immune tolerance and accelerate tumor growth, it is unclear how treatment with adiponectin-expressing EGFP+ cells suppress tumor development.

Answer: We thank the reviewer's comments. Please kindly refer to our responses to **Question 4** above.

13) Most information would be helpful to better understand the data shown in Fig. 8. For example, what is the percentage of tumor-killing IFN- γ ⁺ Th1 and CD8 T cells in the tumor? Is there a functional difference between the two isoforms of CD100? Does the interaction between CD100 and adiponectin or galectin-3 play distinct roles in regulating Treg cell biology and function? How adiponectin regulates the expression of distribution of CD100?

Answer: We appreciate the reviewer's suggestions. Accordingly, the detailed explanation is included in the 4th paragraph of the revised discussion. Please also refer the the above responses to **Question 6**.

14) The authors concluded that adiponectin plays a role in establishing central tolerance by modulating CD100 expression and distribution. However, the data were correlative and it is unclear whether adiponectin plays a distinct role in CD100 KO thymus. Thus, a less strong tone of the conclusion may be more appropriate.

Answer: We thank the reviewer's suggestions. The discussion has been revised with more details for explanation and clarification. We believe that the regulation of CD100 by adiponectin represents at least one of the mechanisms contributing to the establishment of central tolerance. Our current data suggest there are no differences on the expression of AdipoR1 and AdipoR2, as well as T-cadherin of EGFP⁺ cells derived from WT or AKO mice.

Reviewer #2

15) The authors used EGFP⁺ cells as adiponectin-expressing cells in adiponectin-cre/ROSA^{mT/mG} mouse model. However, due to irreversible changes mediated by cre-loxP system, it is elusive whether EGFP⁺ cells consistently express adiponectin or not. To address this issue, adiponectin expression level should be tested in EGFP⁺ thymocytes compared to EGFP⁻ ones. Also, adiponectin protein level in the thymus of EGFP⁺ cell-injected AKO mice should be compared with that of WT counterparts. These experiments would be the key prerequisite step to understand the novel functions of adiponectin-expressing thymocytes.

Answer: We thank the reviewer's suggestions. Accordingly, additional experiment was performed and the results presented in **Supplementary Figure 1** and **Supplementary Figure 2**. Firstly, EGFP⁺ and EGFP⁻ cells were sorted from thymus of Adn-Cre/ROSA^{mT/mG} mice. *ADIPOQ* expression was detected by RT-PCR, whereas adiponectin protein was analyzed by SDS-PAGE and detected by Western blotting. The results confirm that adiponectin is expressed in EGFP⁺ cells (**Supplementary Figure 2**). Indeed, the amount and distribution of adiponectin protein in TNC complexes of AKO recipient are very limited when compared to the WT counterparts (**Figure 3A**).

16). In Figure 4 and 5, administration of EGFP⁺ cells showed dramatic improvement of metabolic parameters in adipose tissue and liver. However, it is still unknown whether these effects would be mediated by T cell development in thymus or directly occurred in adipose tissue and liver. To clarify this issue, it is necessary to show how many EGFP⁺ thymocytes were infiltrated into above organs and to examine the relative ratio of EGFP⁺ cells among resident-Treg cells. Also, it should be investigated the degree of changes in the amount of adiponectin in above tissues (e.g., thymus, adipose tissue, and liver) before and

after EGFP+ cell injection. These experiments could help to explain the mechanism by which EGFP+ thymocytes regulate whole body metabolism.

Answer: Please refer to our response to the above **Question 2** and **Question 7**.

17) The authors claimed that most of EGFP+ cells were colocalized with TNCs in thymus. However, there is no direct evidence supporting this statement because they only showed confocal images of single TNC, but not that of other regions such as medullar regions. In Figure 1B, adiponectin signal seems to be found in medullar regions. Thus, it is required to provide additional tissue section images that show EGFP signal and TNC markers (e.g., CK5, CK8) in overall thymus, including TNC and other regions. Moreover, quantitation data of colocalization signals would be helpful to support their model/proposal. EGFP+ CK5 CK8 after injection and col-local.

Answer: We thank the reviewers' suggestions. The experiment was performed and results included in **Figure 1** and **Figure 2**. As adiponectin is expressed in T-lineage cells enclosed by the epithelial TNC complexes, it should not co-localize with epithelial markers in images acquired under higher magnification, such as those in **Figure 1D** and **Supporting Figure 4**. Please also refer to our responses to **Question 3** above.

18) The current study includes several western blot data. However, some of them need to be improved. For example, in Figure 1A, adiponectin band was cropped. Also, in Figure 8B, there are no positive markers that verify the credibility of TNC fractionation. It is needed to provide full blot images of western blots and conduct additional western blotting to show successful fractionation of TNC.

Answer: We thank the reviewer's suggestions. The full blot images are included in **Figure 1**. Other full-size gel image will be sent to editor upon request. The positive markers of TNC, β 5t, was included in **Figure 8B** to verify the credibility of TNC fractionation. In addition, images of enriched TNC are included in **Figure 2C** and **Figure 7D**.

Reviewers' comments:

Reviewer #1 (Remarks to the Author):

The authors have carefully revised the manuscript and adequately addressed most of my critiques. However, there are a number of questions remain un-answered and answers to these questions would greatly improve the manuscript.

6) Are adiponectin receptors and other signaling components highly expressed in Treg cells?

9) Are adiponectin expression and oligomerization in the thymus altered under pathophysiological conditions such as obesity?

10) Since the donor cells were the same. it is unclear whether the different microenvironment of recipient WT and AKO mice affected the donor cell development.

The authors have acknowledged that different microenvironment of the recipient WT and AKO mice affected the donor cell development. However, their argument that “Note that in TNC complexes of AKO recipient mice, the adiponectin protein produced by the EGFP+ cells is very limited when compared to those in the WT TNC complexes (Figure 3A).” This statement is incorrect given that adiponectin protein produced by the EGFP+ cells is comparable (Fig. 3A, green) regardless of the recipient mice. Only the adiponectin protein produced by the TNC cells of AKO recipient mice is limited when compared to those by the TNC complexes of WT recipients (Fig. 3A, red).

10) It is also unclear why the adiponectin-expressing donor cells could not support their own development in the AKO recipient mice?

13) what is the percentage of tumor-killing IFN- γ + Th1 and CD8 T cells in the tumor? How adiponectin regulates the expression of distribution of CD100?

The authors only showed how adiponectin regulates the distribution of CD100, but did not discuss how adiponectin regulates CD100 expression. Also, the discussion of the functional differences between the two isoforms of CD100 and the interaction between CD100 and adiponectin or galectin-3 in regulating Treg cell biology and function should be moved forward to the results section to make the background clearer for readers.

Reviewer #2 (Remarks to the Author):

In this study, the authors demonstrated that adiponectin-expressing cells in thymus are differentiated into Treg cells and improve HFD-induced glucose intolerance, liver steatosis, and breast tumor growth. In the previous manuscript, there were several major issues to be solved in revised manuscript. For example, it was not clear whether EGFP+ cells could express adiponectin. Also, the mechanism by which adiponectin modulates T cell selection in TNC complexes was not fully elucidated. In this revised manuscript, they successfully solved the issues raised by the reviewers through several experiments. Also, most experimental results were adequately described.

Point-to-point responses:

Reviewer #1 (Remarks to the Author):

The authors have carefully revised the manuscript and adequately addressed most of my critiques. However, there are a number of questions remain un-answered and answers to these questions would greatly improve the manuscript.

6) Are adiponectin receptors and other signaling components highly expressed in Treg cells?

Answers: We thank the reviewer's comments. As shown in the below QPCR results (**Figure S1**), the AdipoR1 and AdipoR2 are not highly expressed in CD4⁺CD25⁺ Treg cells when compared to those of thymus. T-cadherin, another adiponectin receptor, is highly expressed in TNC complexes. Nevertheless, as it is unclear whether or not these receptors are involved in T-cell selection and development, we prefer not to include the data in the revised manuscript.

Figure S1: Thymus was collected from seven-weeks old wild type mice to isolate CD4⁺CD25⁺, CD4⁺CD25⁻ and TNC complexes for RNA extraction and QPCR analyses. The mRNA expression levels of genes including *ADIPOR1*, *ADIPOR2* and *CDH13* were measured for comparison. Data are presented as mean \pm SEM. *, $P < 0.05$ and **, $P < 0.01$ vs thymus (n=3).

9) Are adiponectin expression and oligomerization in the thymus altered under pathophysiological conditions such as obesity?

Answers: We thank the reviewer's question. Accordingly, we have performed the experiment to compare the oligomers of adiponectin in thymus of MMTV-PyVT mice, which express the transgene encoding mouse mammary tumor virus (MMTV) long terminal repeat upstream of polyomavirus middle T oncogene (PyVT) and spontaneously develop mammary tumors, and C57BL/6J wild type (WT) mice under standard chow (STC) or high fat diet (HFD) feeding. The results in **Figure S2** show significantly decreased amounts of total adiponectin in thymus of 14-weeks old MMTV-PyVT transgenic (PyVT^{+/-}) or WT mice subjected to 15-weeks of HFD, when compared to PyVT^{-/-} or age-matched WT under STC, respectively. However, there are no significant differences in the ratios between HMW and hexameric adiponectin. In addition, the oligomers within the TNC complexes were also analyzed and compared to thymus (**Supplementary Figure 2B; descriptions in 3rd paragraph of page 6**).

Figure S2: Thymus tissues were collected from MMTV-PyVT mice without (PyVT^{-/-}) or with (PyVT^{+/-}) the PyVT transgene, and WT mice under STC or HFD. Same amount of tissue lysates (30 μ g) were separated by non-reducing or denatured SDS-PAGE for analyzing the oligomers and total amount of adiponectin for comparison. Beta-actin (β -actin) was probed as loading controls. Data are presented as mean \pm SEM. *, $P < 0.05$ vs PyVT^{-/-} or STC (n=6).

10) Since the donor cells were the same, it is unclear whether the different microenvironment of recipient WT and AKO mice affected the donor cell development.

Answers: Since the donor cells were the same, it is highly possible that the different microenvironment in TNC complexes of recipient WT and AKO affected the donor cell development. To test this hypothesis, we performed preliminary analyses of the TNC complexes isolated from thymus of WT and AKO mice using the Seahorse XFe24 analyzer. To this end, TNC complexes containing the same number of total cells were subjected to the measurement of oxygen consumption rate (OCR) and extracellular acidification rate (ECAR) in a multi-well plate. The results in **Figure S3** demonstrate that TNC complexes collected from AKO mice are less active in energy metabolism, no matter oxidative phosphorylation or aerobic glycolysis. Although the data suggest that the micro-metabolic environment of TNC complexes is different between WT and AKO samples, it is not clear whether or not the differences are directly related to the presence or absence of adiponectin, or the development of adiponectin-expressing Treg. We are carrying on further experiment to investigate the role of adiponectin in regulating the microenvironment of TNC complexes.

Figure S3: TNC complexes were isolated from seven weeks old wild type (WT) and adiponectin knockout (AKO) mice. Samples containing a total amount of 5×10^5 thymocytes were seeded in the 24-well Agilent seahorse XF microplate. Mitochondrial and glycolysis stress assays were performed according to manufacturer's instructions. Data were presented as mean \pm SEM. *, $P < 0.05$ vs WT TNC samples (n=4).

The authors have acknowledged that different microenvironment of the recipient WT and AKO

mice affected the donor cell development. However, their argument that “Note that in TNC complexes of AKO recipient mice, the adiponectin protein produced by the EGFP+ cells is very limited when compared to those in the WT TNC complexes (Figure 3A).” This statement is incorrect given that adiponectin protein produced by the EGFP+ cells is comparable (Fig. 3A, green) regardless of the recipient mice. Only the adiponectin protein produced by the TNC cells of AKO recipient mice is limited when compared to those by the TNC complexes of WT recipients (Fig. 3A, red).

Answers: In **Figure 3A**, the green signals are derived from EGFP protein but not adiponectin, whereas the red signals represent adiponectin protein expressed from both the EGFP+ cells of the donor mice and TNC complexes of the recipient mice. However, there was no adiponectin produced from TNC complexes of AKO recipient mice, which may contribute to the significantly reduced adiponectin molecules. The results in **Figure 3A**, **Figure 4** and **Figure 9** collectively suggest that both the total amount (e.g. concentration-dependent regulation of the microenvironment) and the location/distribution (e.g. protein-protein interactions) of adiponectin play a role in the development Treg and T-cells within the TNC complexes.

10) It is also unclear why the adiponectin-expressing donor cells could not support their own development in the AKO recipient mice?

Answers: Apart from the quantity or distribution of adiponectin mentioned above, we think there may exist a feedback negative regulation that needs to be further investigated. For example, since the microenvironment of the AKO TNC complexes is already different, after adoptive transfer, the adiponectin-expressing Treg could not develop efficiently and maintain the expression of CD25 for cell proliferation. The reduced number of cells would lead to a further reduction of adiponectin expression when compared to WT recipient mice. In fact, based on the number of EGFP+ cells in TNC complexes after 15 days of adoptive transfer (**Figure 3D**), the total amount of adiponectin expressed from the EGFP+ donor cells in the TNC complexes of AKO mice would be at least three-fold less than that of WT recipient mice, in turn affecting the TNC microenvironment as well as T-cell and Treg development. In the meantime, the altered microenvironment precludes the normal function of adiponectin-expressing Treg precursors or even adiponectin expression and production from these cells. We are testing these hypotheses at the moment.

13) what is the percentage of tumor-killing IFN- γ + Th1 and CD8 T cells in the tumor? How adiponectin regulates the expression of distribution of CD100?

Answers: We thank the reviewer's suggestions. In the revised manuscript, the results of CD4+ and CD8+ cells in tumors collected from MMTV-PyVT mice treated with vehicle or adiponectin-expressing EGFP+ cells were included (**Supplementary Figure 6A**; descriptions in **the 1st paragraph of page 11**). Both types of cells significantly increased in tumors of MMTV-PyVT mice treated with adiponectin-expressing EGFP+ cells. The results are consistent with CD3 staining in tumor samples (**Figure S4**). Treatment with adiponectin-expressing EGFP+ cells significantly increased the amount of CD3+ cells infiltrated within the tumor tissues, but reduced the number of CD100+ cells. Interestingly, about 89% of CD8+ cells are also CD100+, but CD3-, in MMTV-PyVT tumors. At this stage, we do not know what are these cells and how

they affect tumor development, which will need to be thoroughly investigated, but is beyond the scope of the present study. Similarly, IFN- γ is not only expressed by Th1 but also by CD8+ T-cells and Treg. In fact, the role of IFN- γ in tumor development is not conclusive. Both pro- and anti-tumorigenic effects of IFN- γ have been demonstrated (Jorgovanovic D et al Biomark Res. 2020 Sep 29;8:49; Burke JD and Young HA Semin Immunol. 2019 Jun;43:101280). To avoid the diversion of our focus, we prefer not to analyze the IFN- γ + Th1 cells at this stage in this manuscript. We will certainly consider the suggestions in our ongoing project focusing on the immune-microenvironment of the tumors.

Figure S4: Vehicle or EGFP+ cells [30000 cells/per mouse] collected from the thymus of four to five-weeks old Adn-Cre/ROSA^{mt/mg} were injected into four-weeks old MMTV-PyVT mice via tail vein. At the age of 14-weeks, mice were sacrificed to collect tumor samples for analysis. Paraffin sections prepared from tumor tissues were stained with antibodies against CD3, CD8 or CD100 (recognizing the region of amino acid 24-657). All slides were counterstained with DAPI. CD3⁺, CD100⁺ and CD8⁺CD100⁺ cell numbers were manually counted for comparison. Data were presented as mean \pm SEM. *, $P < 0.05$ vs corresponding controls (n=6).

The authors only showed how adiponectin regulates the distribution of CD100, but did not discuss how adiponectin regulates CD100 expression. Also, the discussion of the functional differences between the two isoforms of CD100 and the interaction between CD100 and adiponectin or galectin-3 in regulating Treg cell biology and function should be moved forward to the results section to make the background clearer for readers.

Answers: The presence or absence of adiponectin does not affect the mRNA expression of CD100. Adiponectin modulates the protein expression levels of CD100 (**Figure 9B**) mainly at the posttranslational level by inhibiting the proteolytic cleavage (**descriptions in page 14 and discussions in the last paragraph of page 17**). However, the detailed mechanisms are not clear. We are exploring the possibilities of the involvement of different metalloproteinases (Rajabinejad M et al Gene. 2020 Jul 1;746:144637) or intracellular signaling pathways (Mou P et al Blood. 2013 May 16;121(20):4221-30; Chen T et al Platelets. 2016 Nov;27(7):673-679). The background information of galectin-3 is included in the results (**2nd paragraph of page 8**).

Reviewer #2 (Remarks to the Author):

In this study, the authors demonstrated that adiponectin-expressing cells in thymus are differentiated into Treg cells and improve HFD-induced glucose intolerance, liver steatosis, and

breast tumor growth. In the previous manuscript, there were several major issues to be solved in revised manuscript. For example, it was not clear whether EGFP+ cells could express adiponectin. Also, the mechanism by which adiponectin modulates T cell selection in TNC complexes was not fully elucidated. In this revised manuscript, they successfully solved the issues raised by the reviewers through several experiments. Also, most experimental results were adequately described.

Answer: We thank the reviewer's positive comments and have no further responses.

REVIEWERS' COMMENTS:

Reviewer #1 (Remarks to the Author):

The authors have adequately addressed my questions and the manuscript is now suitable for publication in the journal.

Reviewer #2 (Remarks to the Author):

This reviewer has no further comments.